# WRN structural flexibility showcased through fragment-based lead discovery of inhibitors

Rachel L. Palte[1,9], Mihir Mandal[2,9], Justyna Sikorska ®[3,9], Artjohn B. Villafania[3,9], Meredith M. Rickard ®[1], Robert J. Bauer[4], Alexei V. Buevich[5], Xiaomei Chai[4], Jiafang He[2], Zahid Hussain[2], Markus Koglin[3], Hannah B. MacDonald[6], My S. Mansueto[4], Klaus Maskos[7], Joey L. Methot[1], Jaclyn Robustelli[3], Aileen Soriano[3], Marcel J. Tauchert ®[7], Sriram Tyagarajan[2], Minjia Zhang ®[4], Daniel J. Klein[8], Jacqueline D. Hicks[2], David G. McLaren[3], Sandra B. Gabelli ®[8] ✉ & Daniel F. Wyss ®[3] ✉

WRN helicase is an established synthetic lethal target for inhibition in the treatment of microsatellite instability-high (MSI-H) and mismatch repair deficient (MMRd) cancers. The identification of helicase inhibitors is challenging as high-throughput biochemical screening campaigns typically return few validated hits that are often inactive in cell-based assays. Herein, we highlight the power of non-covalent fragment-based lead discovery in locating new druggable allosteric sites on WRN, enabling us to bypass the challenging behavior of WRN during high-throughput screening hampering hit identification. During the fragment optimization process, structures of WRN with key prioritized fragments reveal multiple conformations of WRN with significant domain rotations up to 180°, including a WRN conformation not previously described. Rooted in a combination of biochemical, biophysical, and structural approaches, we present the detailed analyses of optimized chemical matter evolved from screening hits and the unique ability of WRN to accommodate diverse conformations as detailed by structural characterization.

Synthetic lethal vulnerabilities have rapidly gained interest as therapeutic targets for cancer. A term coined in 1946[1], synthetic lethality occurs when disruptions in two genes cause cell or organism death, but single mutations in either of these genes spare the cell or organism. The ability to selectively inhibit targets of interest while minimizing deleterious effects on healthy cells has led to the development of new therapies to treat cancer[2,3]. Notable examples of this type of drug therapy are PARP inhibitors for the treatment of cancers with mutations in BRCA1/2, which are DNA damage repair proteins[4,5].

Cancer cells often exhibit at least partial impairment of specific DNA repair pathways, which may cause dependence on the actions of distinct repair proteins for cell survival[6]. Werner syndrome helicase (WRN) is known as a guardian of genome integrity, due to its critical roles in numerous aspects of nucleic acid metabolism, including genome surveillance, maintenance, and stability[7]. Multiple studies identified a synthetic lethal relationship between loss of WRN and microsatellite instability (MSI), which arises from deficiencies in the mismatch repair (MMR) machinery[8–10]. These studies showed that

[1]Discovery Chemistry, Merck & Co., Inc., MRL, Boston, MA, USA. [2]Discovery Chemistry, Merck & Co., Inc., MRL, Rahway, NJ, USA. [3]Quantitative Biosciences, Merck & Co., Inc., MRL, Rahway, NJ, USA. [4]Quantitative Biosciences, Merck & Co., Inc., MRL, Boston, MA, USA. [5]Analytical Research & Development, Merck & Co., Inc, Rahway, NJ, USA. [6]Discovery Chemistry London, MSD (UK) Ltd, London, UK. [7]Proteros Biostructures, Planegg-Martinsried, Martinsried, Germany. [8]Discovery Chemistry, Merck & Co., Inc., MRL, West Point, PA, USA. [9]These authors contributed equally: Rachel L. Palte, Mihir Mandal, Justyna Sikorska, Artjohn B. Villafania. ✉e-mail: Sandra.gabelli@merck.com; Daniel.wyss@merck.com

WRN is essential in MSI-high (MSI-H) cells, which are MMR-deficient, as the depletion of WRN induces double-stranded DNA breaks and promotes apoptosis and cell cycle arrest[9,10]. In contrast, deletion of WRN in microsatellite stable (MSS) cells shows no adverse events. This reliance on WRN by MMRd cells suggests a high therapeutic index for agents that disrupt WRN function in the treatment of MSI-H tumors.

WRN belongs to the five-member RecQ family of super family 2 (SF2) helicases, including RECQL1, Bloom syndrome protein (BLM), RECQL4, and RECQL5, all of which unwind DNA in a 3′−5′ direction[11]. Despite their DNA substrate dependence, the distributive RecQ helicase core has significant structural similarities to DEAD-Box RNA helicases rather than processive DNA replicative helicases[12,13]. Each RecQ family member features a helicase core domain consisting of two RecA-like ATPase subdomains, D1 and D2. Additionally, WRN contains a RecQ C-terminal (RQC) domain that consists of a winged helix (WH) domain, a zinc-binding domain (ZBD), and an HRDC (helicase and RNaseD C-terminal) domain[11]. Notably, WRN – along with its Xenopus ortholog FFA-1[14]−is distinct within the RecQ family as the only protein that possesses an N-terminal exonuclease domain exhibiting 3′−5′ exonuclease activity[15].

Helicases, in general, have proven to be challenging drug targets. Common problems include a lack of protein specificity arising from the highly conserved ATP pocket, low hit rates in high-throughput screens in standard in vitro helicase assays, and non-specific binding or activity of identified hits[16]. Various research groups have identified potential WRN inhibitors[17-21], however no successful developments of discovered chemical matter binding the ATP pocket has been reported. This lack of success has recently been thoroughly evaluated by Hauser et al., showing that non-specific interactions can be attributed to impurities in the test samples (ML216) or covalent labeling of the WRN protein (NSC compounds)[22]. In addition to these challenges, the ATP-binding site in helicases is predicted to be less likely to bind a drug-like molecule than enzymes like kinases, complicating the search for potent ATP-competitive inhibitors[23]. These considerations indicate that allosteric inhibition of WRN might facilitate the identification of selective leads among RECQ helicases.

Recently, literature has disclosed the discovery of two conformational states of WRN[24,25]. Prior to those studies, only one WRN helicase structure was available in the Protein Data Bank (PDB: 6YHR)[7] which featured WRN with bound ADP and resembled all other known RecQ family helicase structures. We refer to this as WRN Form A, as it was the first structure to be characterized. In one of these two recent studies[24], a noncovalent compound currently under clinical development was shown to bind to an allosteric site that occupies the cleft between the two ATPase domains, effectively locking WRN in an inactive conformation. This mechanism of action is rationalized by the observation that, in this conformation, the RecQ ATPase domains D1 and D2 are rotated by approximately 180° compared to the ATP-bound WRN. This led us to label this WRN arrangement as Form B. A contemporaneous paper described a third form of WRN (Form C) in which WRN can bind both ADP and a covalent compound in the allosteric pocket simultaneously[25]. This structure displays smaller movements between the ATPase domains compared to Form B but nevertheless stabilizes compact conformations that lack the dynamic flexibility required for proper helicase function[26-30].

Given the challenges of identifying high-quality WRN inhibitors through high-throughput screens based on the biochemical assays formerly described in literature[22], we proposed screening approaches rooted in biophysics, such as fragment-based lead discovery (FBLD) or DNA-Encoded Library (DEL), as the next rational starting points for targeting this enzyme. FBLD workflows are particularly effective in identifying alternative binding sites, albeit often a mix of functional and non-functional ones, and expediting the development of novel chemical matter. These workflows leverage biomolecular Nuclear Magnetic Resonance (bioNMR) and Surface Plasmon Resonance (SPR) techniques to rapidly identify small molecules (molecular weight <300 Da) with low affinities that can be optimized toward more ligand-efficient leads. To navigate this process successfully, it is crucial to concurrently generate structural data that can elucidate the atomic interactions between the fragment hit and the target protein.

In this study, we employ FBLD to identify WRN inhibitors through a combination of [19]F ligand-based NMR and SPR screening of internal fragment libraries. The prioritized hits are further analyzed using X-ray crystallography, which reveals the specific binding mode of these compounds at the known binding site of WRN in Form B, identifies an additional allosteric pocket, and uncovers a previously uncharacterized structural conformation of the WRN helicase domain with unique orientations of the ATPase domains. Ultimately, despite the apparently highly dynamic nature of WRN, which poses challenges to structure-based drug discovery, we advance one of the fragment series to achieve low µM inhibitory activity.

## Results

### Fragment-based screens identify allosteric binders to WRN

The integration of fragment-based screening with structure-based drug design, enabled by crystallography, provides a powerful approach for discovering and developing compounds that bind to small-molecule hotspots on the target's surface[31], including the successful application of fragment screening to identify additional allosteric binding pockets[32]. To identify further binding sites on WRN, we employed our FBLD platform, which relies on screening our internal fragment collections that were a priori curated to only include compounds with sufficient solubility and purity. Given the highly dynamic nature of helicases[26-30], we opted to perform our fragment screen using the apo form of WRN to identify the most druggable allosteric site, avoiding any bias toward ATP-blocked conformations or other potential WRN conformations present in the solution.

We implemented two parallel screening workflows: bioNMR- and SPR-based approaches (Figure SI 1) against WRN (residues 500−942 and 500−946, respectively; Figure SI 2, Table SI1). In the bioNMR studies, we screened an internal collection of fluorine-containing fragments in mixtures using $T_2$ relaxation CPMG experiments to identify primary hits. To address fluorine's heightened sensitivity to changes in the chemical environment, which can lead to an increased false-positive rate, each primary hit underwent orthogonal validation in singleton mode using [19]F $T_2$ CPMG, [1]H $T_2$ CPMG, STD (Saturation-Transfer Difference), and waterLOGSY NMR experiments, enhancing the confidence in the identified fragment hits.

Based on the results (Fig. 1, Figure SI 3), hits were categorized into three ranks: Rank 1 for those showing binding with all four methods, Rank 2 for those showing binding with any three methods, and Rank 3 for those showing binding with two or fewer methods[33]. Each experiment provided insights into different aspects of protein-ligand interaction−STD assessed the transfer of saturation from the protein directly to bound ligands; waterLOGSY measured the transfer of magnetization from excited water molecules directly to the compounds; and [1]H CPMG evaluated the signal attenuation of small molecules bound to a slowly tumbling protein. We hypothesized that compounds binding across all four methods were interacting with the most druggable binding cavity on the protein and prioritized these for crystallization efforts.

Notably, the three fragments that crystallized were bound within the Form B allosteric binding pocket[25,34]. Structure-based chemical expansion of two of those overlapping validated fragment hits (**fragment 1** and **fragment 2**) is discussed. Finally, we conducted a biochemical assessment to evaluate the inhibition of WRN ATPase activity for each compound of interest by measuring ADP accumulation during steady-state turnover (using the ADP-Glo assay) in the presence of 1X $K_m$ ATP (Table SI 2, Figures SI 2a, 4). However, no parental fragments

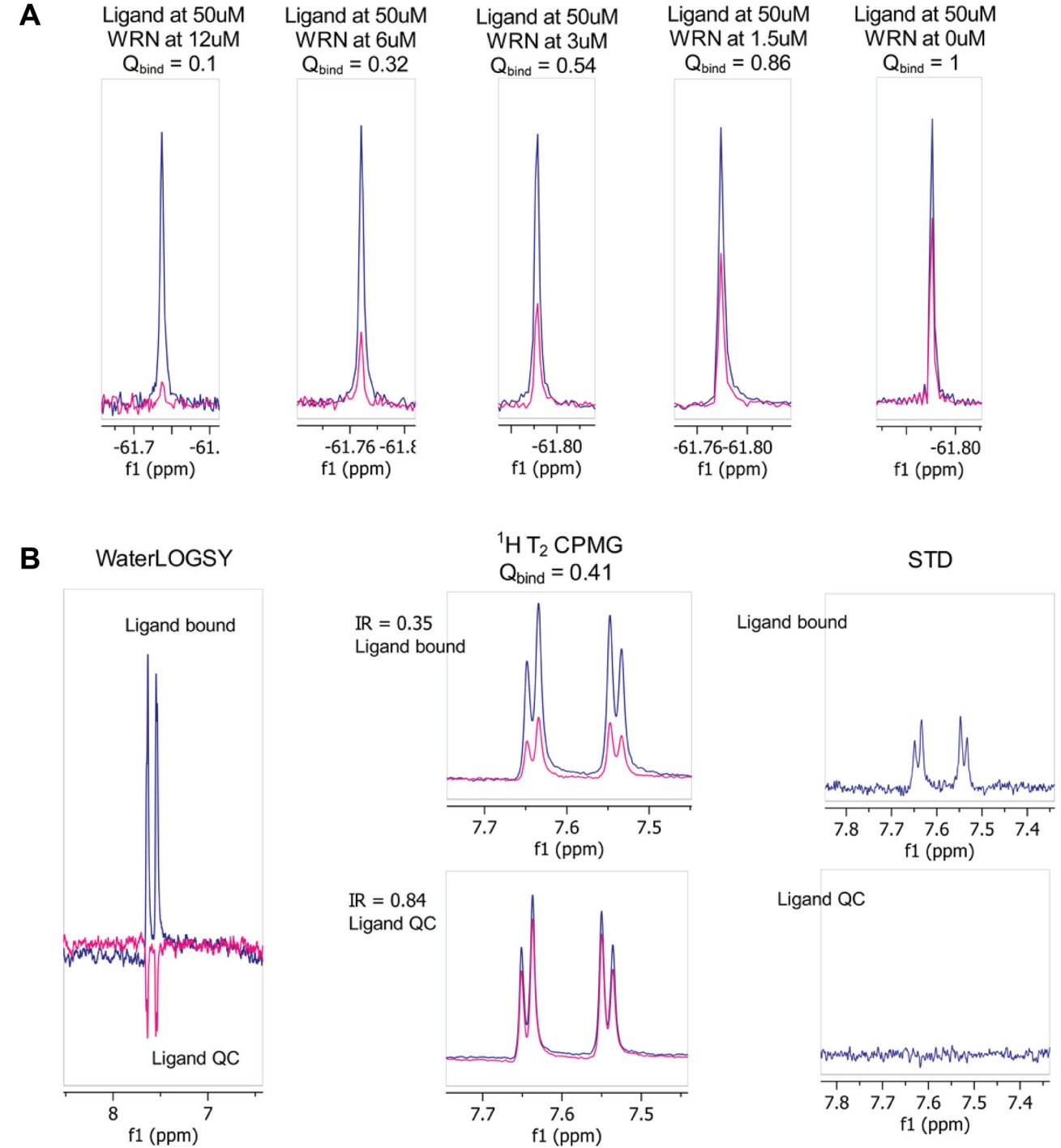

**Fig. 1 | NMR data for fragment 1 used in the prioritization for X-ray crystallography.** All samples were prepared in 25 mM HEPES, 50 mM NaCl, 2 mM MgCl$_2$, pH 7.4 buffer and DMSO-D6 always matched to 2.5% final sample volume. **A** Fragment 1 $^{19}$F T$_2$ CPMG experiment confirming dose-dependent interaction with WRN. **B** Fragment 1 binding to WRN in the water-LOGSY, $^1$H T$_2$ CPMG and STD experiments, placing it among the Rank 1 hit compounds.

were able to significantly inhibit WRN, even at compound concentrations up to 100 μM.

To further investigate the chemical space accessible through our non-fluorine fragment library, we conducted a screen using an SPR binding assay performed with the apo form of WRN. A selected set of 500 fragments was tested in a single-concentration format (1 mM), and each fragment was ranked according to its binding occupancy level and binding profile (Fig. 2). The top hits from the SPR assay (Rank 1) were further evaluated with a 5-point concentration response assay and subjected to orthogonal validation in

singleton mode using $^1$H T$_2$ CPMG, STD, waterLOGSY NMR, and biochemical ATPase experiments.

Since none of the parental fragments significantly inhibited WRN at concentrations up to 100 μM (Table SI 2), we intentionally selected compounds exhibiting either Rank 1 or Rank 3 binding profiles from the ligand-observed NMR experiments. While Rank 1 compounds provided the highest confidence in their interactions with the protein, fragments with differential binding profiles could still represent putative binders interacting with WRN in different ways. Notably, two compounds from the SPR screen yielded X-ray

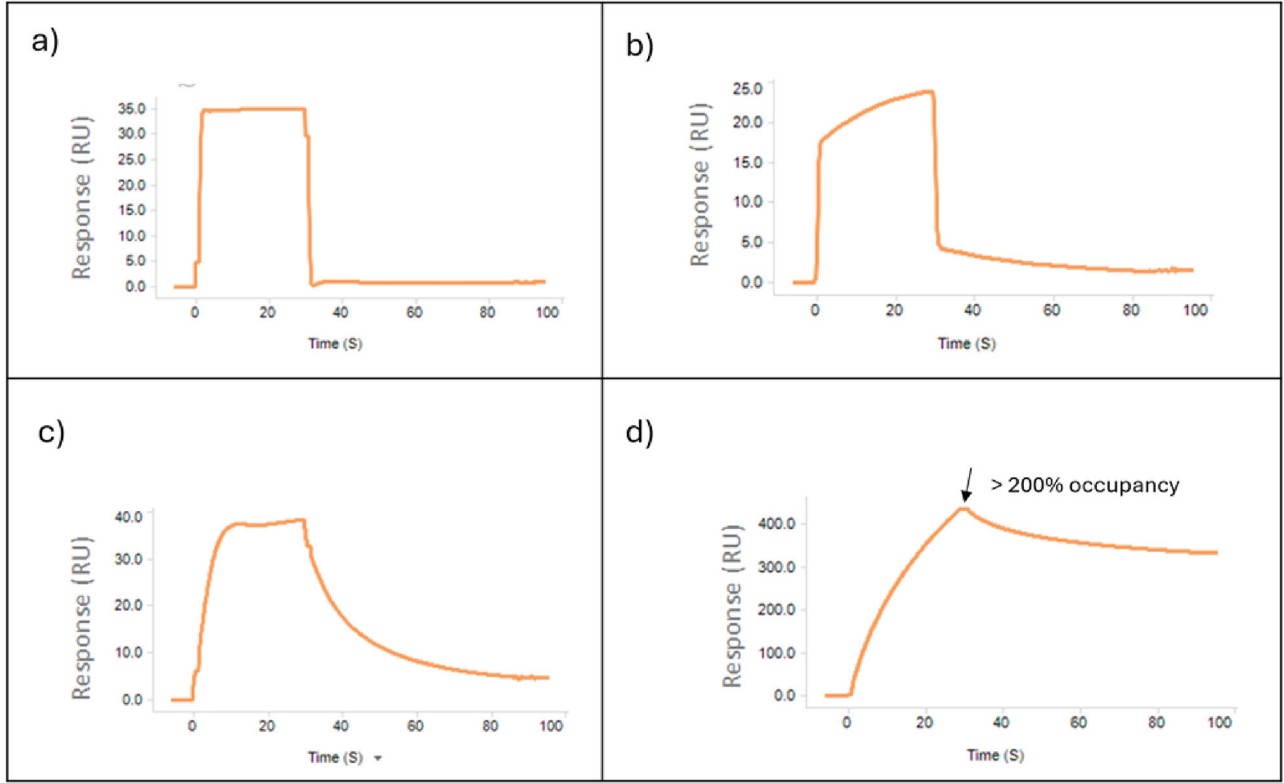

**Fig. 2 | Prioritization of SPR fragment screening hits based on binding profile.** The prioritization of fragment hits was carried out by clustering each test fragment based on binding sensorgram characteristics. Each compound cluster was ranked as follows: **a** *Rank 1*, fast-on, fast-off binding kinetics; **b** *Rank 2*, biphasic binding profile with predominantly fast-on, fast-off binding and a minor component (roughly ≤ 20% of the binding profile) of slow-on, slow-off binding kinetics; **c** *Rank 3*, significant (> 20% of the binding profile) slow kinetic profiles; **d** *Rank 4*, super-stoichiometric/non-saturable binding.

structures: **fragment 3**, which ranked highly in both SPR and NMR, occupied a different binding site, while **fragment 4**, which ranked high in SPR but showed no binding by NMR, occupied the residual ATP binding site. Sensorgrams for these fragments are displayed in Figure SI 5.

Analysis of all identified binding sites led us to prioritize the Form B allosteric binding site and **fragment 1** for further optimization. We hypothesized that the larger volume of the Form B allosteric site would make it more amenable for structure-activity relationship (SAR) exploration as described in more detail below and in Fig. 3.

### Ligand bound structures showcase WRN flexibility and an additional allosteric pocket

For these studies, a truncated construct of WRN (residues 500–942; Figure SI 2A) facilitated a reproducible crystallization system. This construct encompasses the two ATPase domains—D1 and D2—along with the zinc-binding domain (ZBD). Our initial structure of wild-type human WRN (500–942) complexed with AMP-PNP exhibits a root mean square deviation (RMSD) of 0.52 Å (Cα backbone atoms)[24] when aligned with PDB ID 6YHR, a longer WRN construct consisting of residues 517–1093[7], despite lacking the HDRC domain (Figure SI 6). Additionally, both recent papers on WRN drug discovery utilized truncated WRN constructs—WRN (517-946) with six point mutations[24] and wild type WRN (517–945)[25]—to identify and develop inhibitors that showed both in vivo and cellular potency. Together, this data provides confidence that the WRN (500–942) construct is appropriate for structure-based drug design and feasible for structure-activity relationship (SAR) studies. All Fo-Fc omit map ligand densities are shown in Fig. 4.

The structure of WRN with bound **fragment 1** was determined to 1.6 Å resolution revealing that **fragment 1** binds to the recently identified allosteric pocket where the two ATPase domains—D1 and D2—are rotated 180° relative to one another (Fig. 5A). Additionally, a loop composed of residues Val570-Ser578 obstructs the ATP binding pocket (Fig. 5B). Upon closer examination, we determined that while our structure with **fragment 1** resembled Form B, it was not identical as the conformations of the D1 and D2 domains shift approximately 9 Å in relation to each other (Fig. 5C; see Discussion). Due to these conformational differences compared to Form B, we have designated this distinct conformation as WRN Form D. **Fragment 1** makes direct interactions with Arg732, Tyr849, and Arg910, with its trifluoromethyl group situated within a hydrophobic pocket (Fig. 5D). **Fragment 2** binds to WRN in this same pocket, with its trifluoro group similarly positioned to interact with Tyr849 and Arg857 (Figs. 5E, F; structure resolution at 2.6 Å).

Notably, Form D can also be obtained in apo form without the presence of any ligand or inhibitor (Figure SI 7). In this apo Form D, there are no significant changes in the main chain atoms of the protein, and the allosteric pocket remains accessible, while the ATP binding pocket is blocked by the movement of the same Val570-Ser578 loop. In fact, the structures of nearly all ligands presented here were obtained by soaking the compound into apo Form D crystals.

A second allosteric pocket was identified in the 2.0 Å resolution structure of WRN Form D complexed with the SPR-discovered **fragment 3** (Fig. 6A). **Fragment 3** binds to a surface cleft that is approximately 21 Å from the **fragment 1** binding site, positioned at the interface of the D1 and D2 domains. The movement of an N-terminal loop composed of residues Asp523-Ala532 forms this pocket (Fig. 6B),

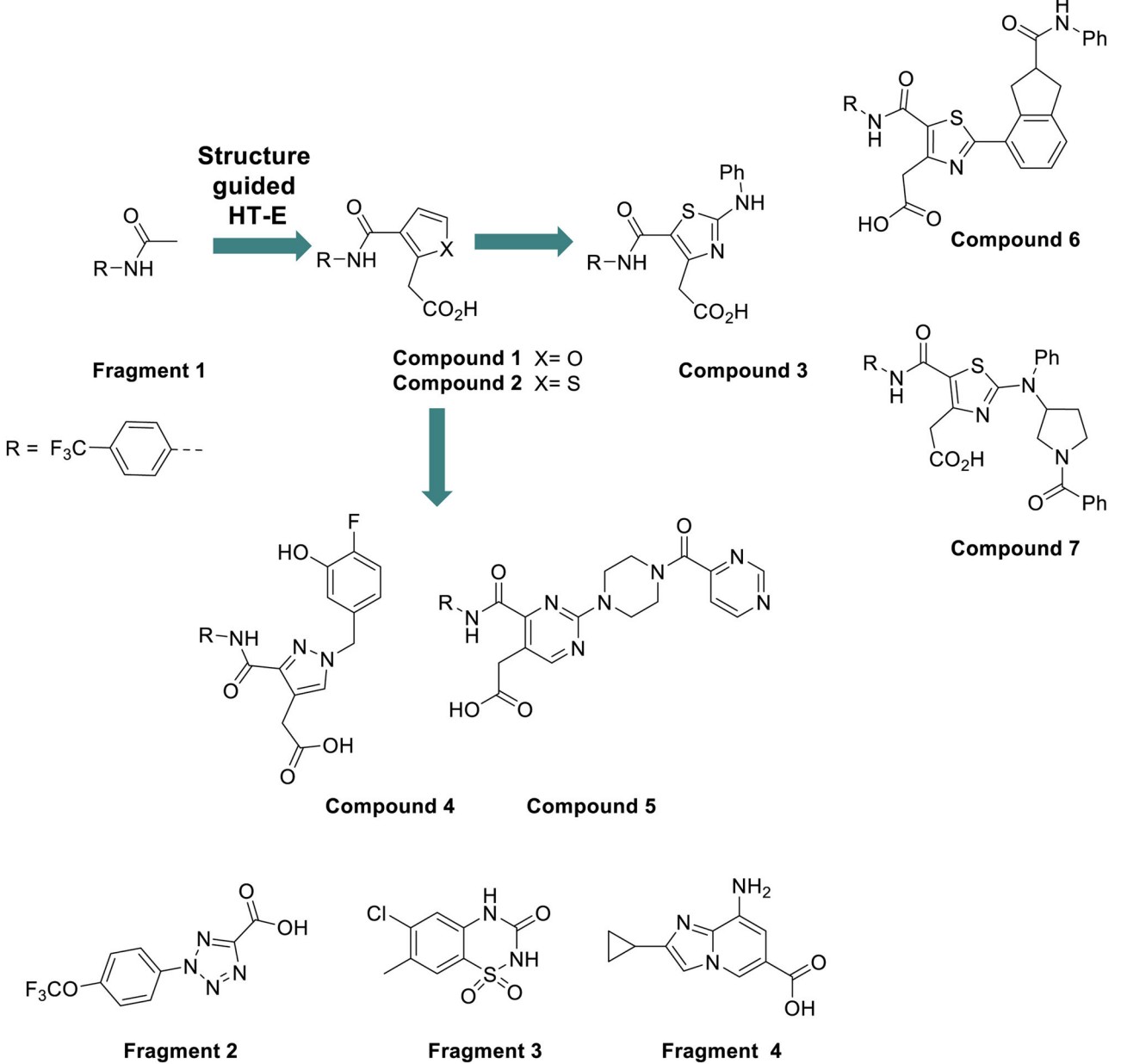

**Fig. 3 | Pathway of fragment-to-lead discovery and development.** WRN inhibitor structures evolved from fragments 1 and 2, highlighting inflection points in the exploration of the series' structure–activity relationships, are presented at the top. Structures of fragments 3 and 4, for which no structure-based chemical expansion was performed, are displayed at the bottom.

where **fragment 3** interacts directly with residues Phe527, Leu528, and Lys887 (Fig. 6C).

Structural studies involving SPR **fragment 4** (1.7 Å resolution) revealed two distinct patches of electron density (Fig. 4) within the remnants of the disrupted ATP binding pocket (Fig. 6D), indicating the presence of two separate molecules at this site. One region of density was confidently assigned to a single molecule of **fragment 4**, while the second region was ambiguous. After analyzing all buffer components and compound byproducts, we chose to model a byproduct of the compound into this space, as it best fit the initial Fo-Fc density (Fig. 6E). Although we cannot determine the exact molecule occupying this second region of density, it is evident that the pocket can accommodate this additional chemical matter. The adjacent loop composed of residues Val570-Lys577, which partially occupies the ATP pocket in Form D, is displaced by approximately 3.5 Å upon the binding of **fragment 4**, aligning more closely with the loop's

conformation when AMP-PNP is bound. Notably, the electron density surrounding the second molecule is somewhat unclear, and the backbone residues of the loop also show faint average electron density, suggesting weak binding of this additional copy and a correspondingly faint average density of the loop. The complete **fragment 4** ligand makes direct interactions with Thr573, Gly574, and Asp668, while the carboxy group of the hypothesized byproduct product interacts with Gln605 (Fig. 6F).

### Structure guided evolution of fragment 1 series leads to functional inhibition and discovery of a WRN Form E

We advanced the development of **fragment 1** by maintaining its interactions with key residues Arg732, Tyr849, and Arg910, while also incorporating insights gained from the X-ray structure of **fragment 2** complexed with WRN. Notably, the carboxylic acid of **fragment 2** forms an important ionic interaction with Arg857, which informed the

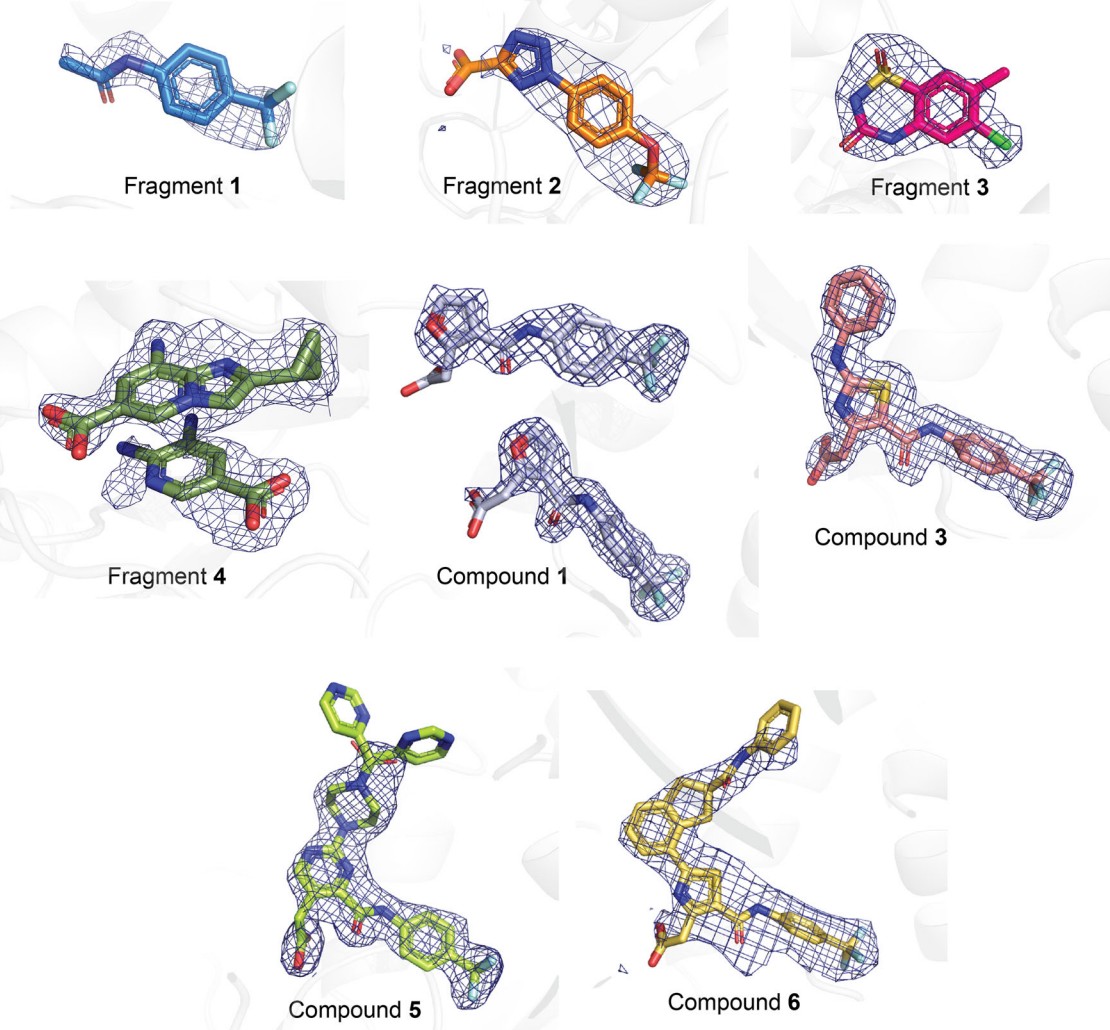

**Fig. 4 | Calculated Fo-Fc electron densities from all ligand-bound X-ray experiments.** All maps are contoured to 2.5 σ. The ligands are colored as follows: **fragment 1** in blue, **fragment 2** in orange, **fragment 3** in pink, **fragment 4** in dark green, **compound 1** in pale blue, **compound 3** in salmon, **compound 5** in lime green, and **compound 6** in yellow.

evolution of **fragment 1**. To design ligands that combined the structural features present in **fragment 1** and **fragment 2**, we generated a virtual library by coupling trifluoroaniline with various proprietary acids from our corporate collection. Virtual ligands were assessed based on their docked poses, focusing on key interactions identified in the crystal structures of WRN with bound **fragment 1** and **fragment 2**. Specifically, we looked for hydrogen bonds with Arg732 from **fragment 1**, hydrogen bonds with Arg857 from **fragment 2**, and π-stacking interactions with Tyr849 observed in both fragments. A total of 17 compounds that satisfied all three criteria were selected for synthesis. Among these, we identified several promising molecules, including **compound** 1 and its thiophene analog, **compound** 2 (Table SI 2, Figure SI 8).

A 1.6 Å resolution X-ray structure of WRN in complex with **compound** 1 revealed that two complete copies of this ligand are bound in the Form D allosteric site (Fig. 4). The trifluorophenyl aniline moiety of the first copy of **compound** 1 aligned well with that of **fragment 1**, displaying a similar H bond interaction with the backbone carbonyl of Tyr849 (Fig. 7A). Additionally, the carboxylic acid group attached to the furan ring via a one-carbon spacer effectively interacts with Arg857, as initially designed. Intriguingly, the trifluorophenyl moiety of the second copy of **compound** 1 was found occupying a hydrophobic

groove formed by Phe917, Tyr849, and Val552. These observations prompted us to undertake several approaches, using different 5 and 6-membered heterocycles and spacers (cf. **3, 4** and **5**) to explore this hydrophobic pocket, using the first copy of **compound** 1 as an anchor.

This investigation resulted in the synthesis of **compound 3**, which exhibited more than a 10-fold improvement in ATPase $IC_{50}$ against WRN, as well as initial evidence of selectivity over BLM (see Table SI 2, Figure SI 8). It was confirmed that **compound 3** binds to the Form D allosteric site. An overlay of the 1.7 Å resolution **compound 3**-bound structure with that of **compound 1** showcases the trajectory and overall binding pattern within this series, with the terminal phenyl occupying a positively charged region of the allosteric pocket (Fig. 7B). The trifluoro moiety of **compound 3** binds in the same anchoring pocket as **fragment 1**, while the remainder of the molecule interacts with nearby protein residues Gln850, Arg732, Tyr849, and Arg857. We additionally observed that Lys555 was in proximity to the terminal phenyl moiety of **compound 3**. Given the flexibility of Lys555, we explored both the meta and para positions of the terminal phenyl of **compound 3**, as well as a previously discovered compound featuring a central pyrazole ring.

From this investigation, we discovered **compound 4**, which exhibited a superior profile within chemical matter made with an $IC_{50}$

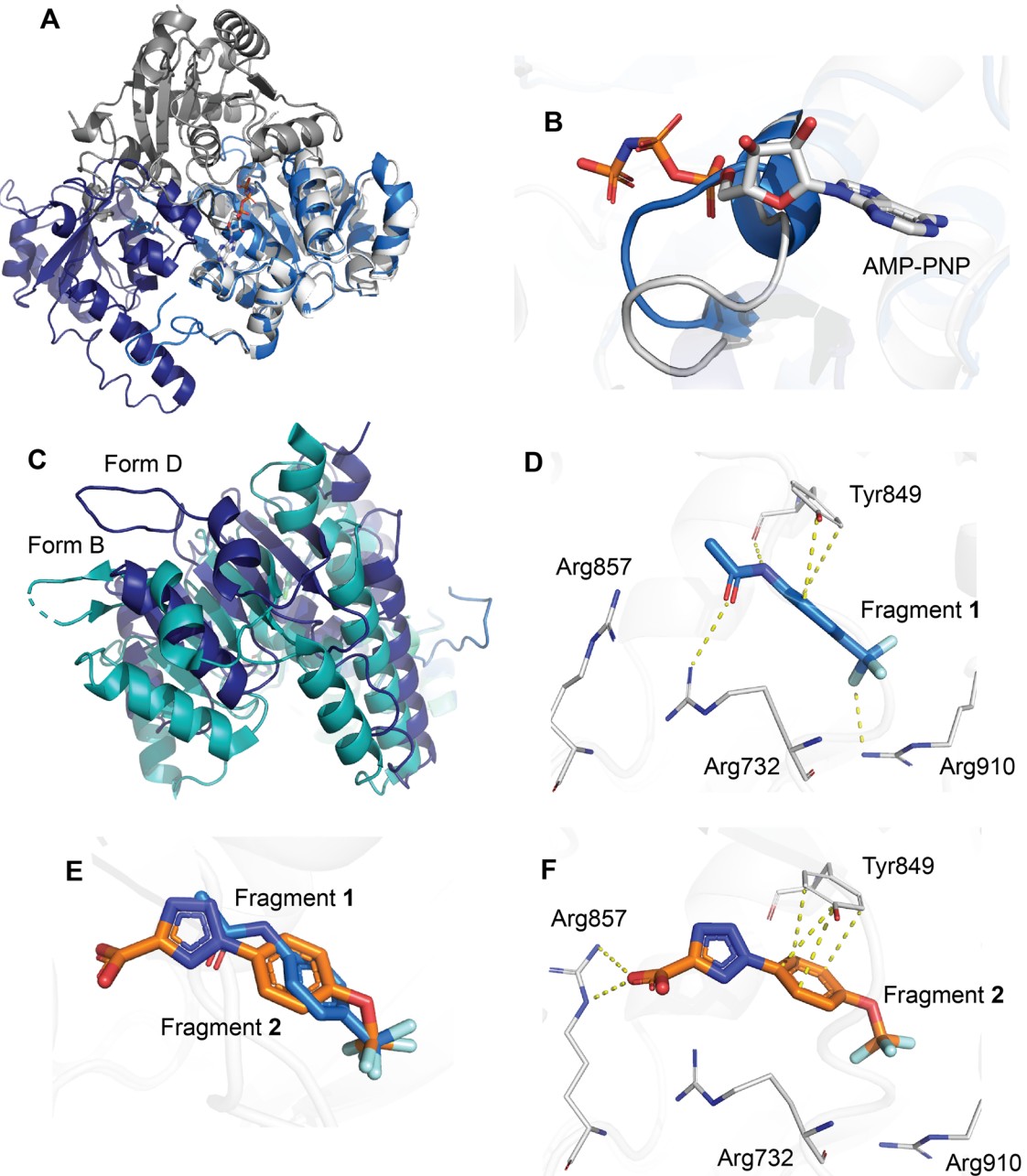

**Fig. 5 | Crystal structures of Form D with ligands of interest. A** Overlay of **fragment 1**-bound WRN (blue) and AMP-PNP-bound WRN (white) structures is aligned on the D1 domains and shows the approximate 180° rotation of the D2 domain. **B** WRN helicase loop consisting of residues Val570-Ser578. This loop closes into and abrogates the binding pocket when **fragment 1** is bound. **C** Structure PDB 8PFO (teal) in Form B superposed with **fragment 1**-bound WRN (dark blue) in Form D. **D** Interactions between **fragment 1** and WRN residues highlighted in yellow dashes. **E** Overlay of **fragment 2** (orange) with **fragment 1** (blue) shows similar ligand binding modes. **F Fragment 2** interactions with WRN residues highlighted by yellow dashes.

of 1.2 μM, a comparable binding affinity of 1.6 ± 0.65 μM, and almost 100-fold selectivity over BLM (Table SI 2, Figure SI 8; Docked pose Figure SI 9). In a parallel effort aimed at forming a hydrogen bond while exploring the binding pocket occupied by the distal phenyl of **compound 3**, we identified both **compounds 6** and **7** (Table SI 2, Figure SI 8). The 2.1 Å resolution X-ray structure of **compound 6** in complex with WRN demonstrated that the portion of the molecule covering the thiazole to the anchoring trifluoro group aligns with **compound 3**, while the extended phenylamide moiety of **compound 6** penetrated into allosteric pocket, surrounded by residues varying in charge states including Pro551, Val552, Lys555, Thr726, Cys727, Glu846, and Phe917 (Fig. 7C). All hydrogen bonds, ionic interactions,

and pi-pi interactions between **compound 3** and WRN are maintained in the binding of **compound 6**, including the interaction to Arg857, with additional pi-pi π-stacking interactions observed between the extended terminal phenylamide in **compound 6** and the side chain of Phe917. To further explore the **compound 3** distal phenyl subpocket, we explored various modifications to the central ring and its substituents. For synthetic ease, we replaced the central thiazole ring of **compound 3** with a pyrimidine moiety and explored various linkers to reach the subpocket of interest. This effort yielded **compound 5**, which was predicted by docking to form a hydrogen bond with Arg854. Figure 3, Table SI 2 and Figure SI 8, Docked pose Figure. SI 9).

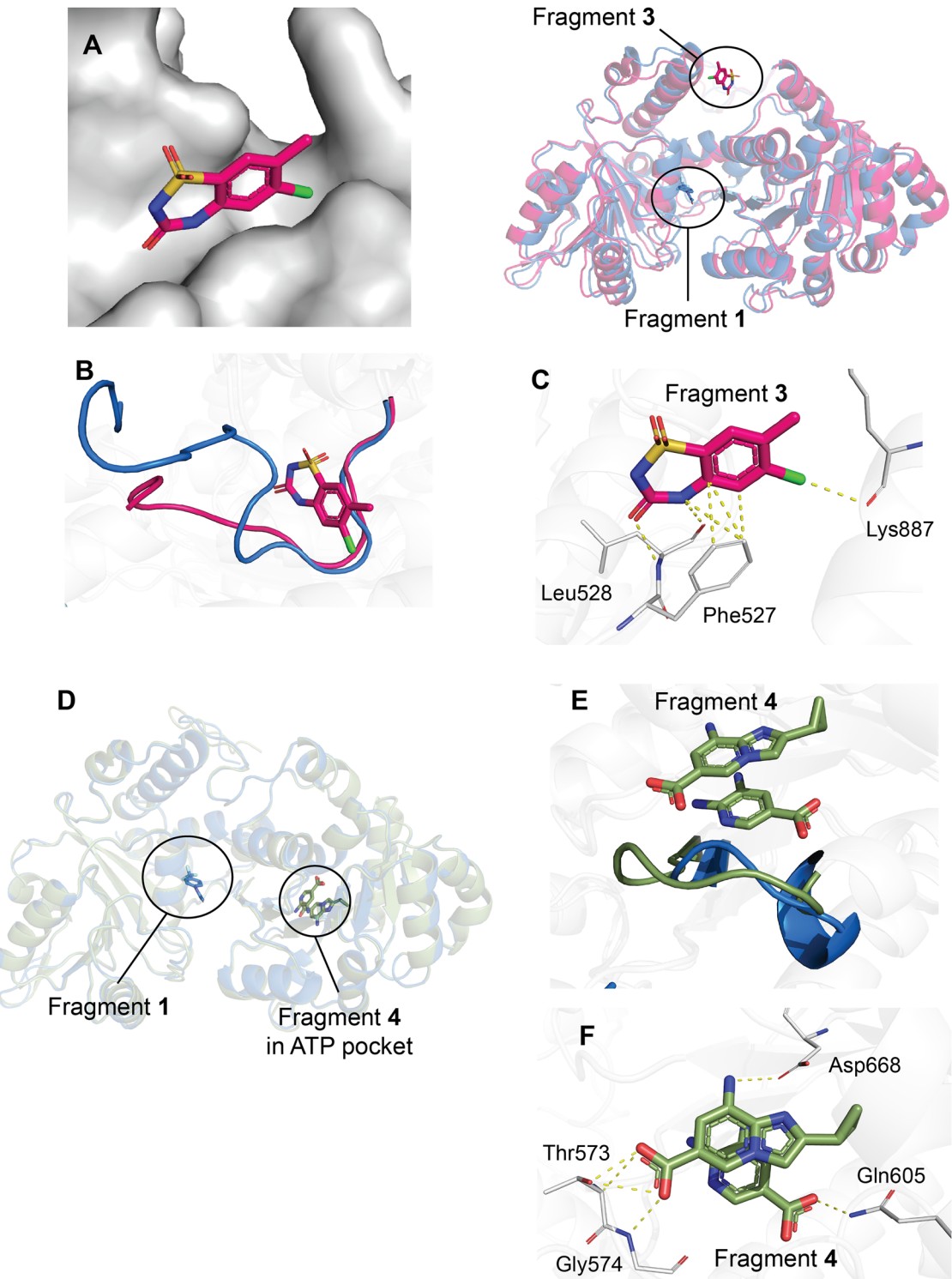

**Fig. 6 | Fragments bound outside of the known allosteric binding pocket.**
**A Fragment 3** (pink) bound to an allosteric pocket on WRN Form D, shown with the
WRN surface in white (left). This **fragment 3**-bound structure is superposed with
**fragment 1**-bound Form D (blue, right). **B** The pocket is formed by the movement
of an approximate 11-residue loop. **C Fragment 3**'s direct interactions with Phe527,
Leu528, and Lys887 are shown in yellow dashes. **D** Two copies of **fragment 4**
(green) bound within the remnants of the ATP pocket in Form D. **E** A loop com-
prising Val570-Lys577 that closes off the ATP pocket in Form D is slightly shifted.
**F** Interactions between nearby residues Thr573, Gly574, Gln605, and Asp668, and
the two molecules seen in the **fragment 4**-bound structure are shown in yellow
dashes.

Remarkably, the structural data for WRN with bound **compound 5**
at 1.9 Å resolution obtained by co-crystallization revealed yet another
conformation of WRN, which we will call WRN Form E (Fig. 8A). This is
notable despite a nearly identical binding mode observed for

**compound 5** and its parent **compound 3**. In Form E, the ATPase
domains exhibit a rotation relative to each other around the linker
region consisting of residues Thr728-Asn734, with the ATP pocket
occluded by the loop comprised of residues Val570-Ser578, as

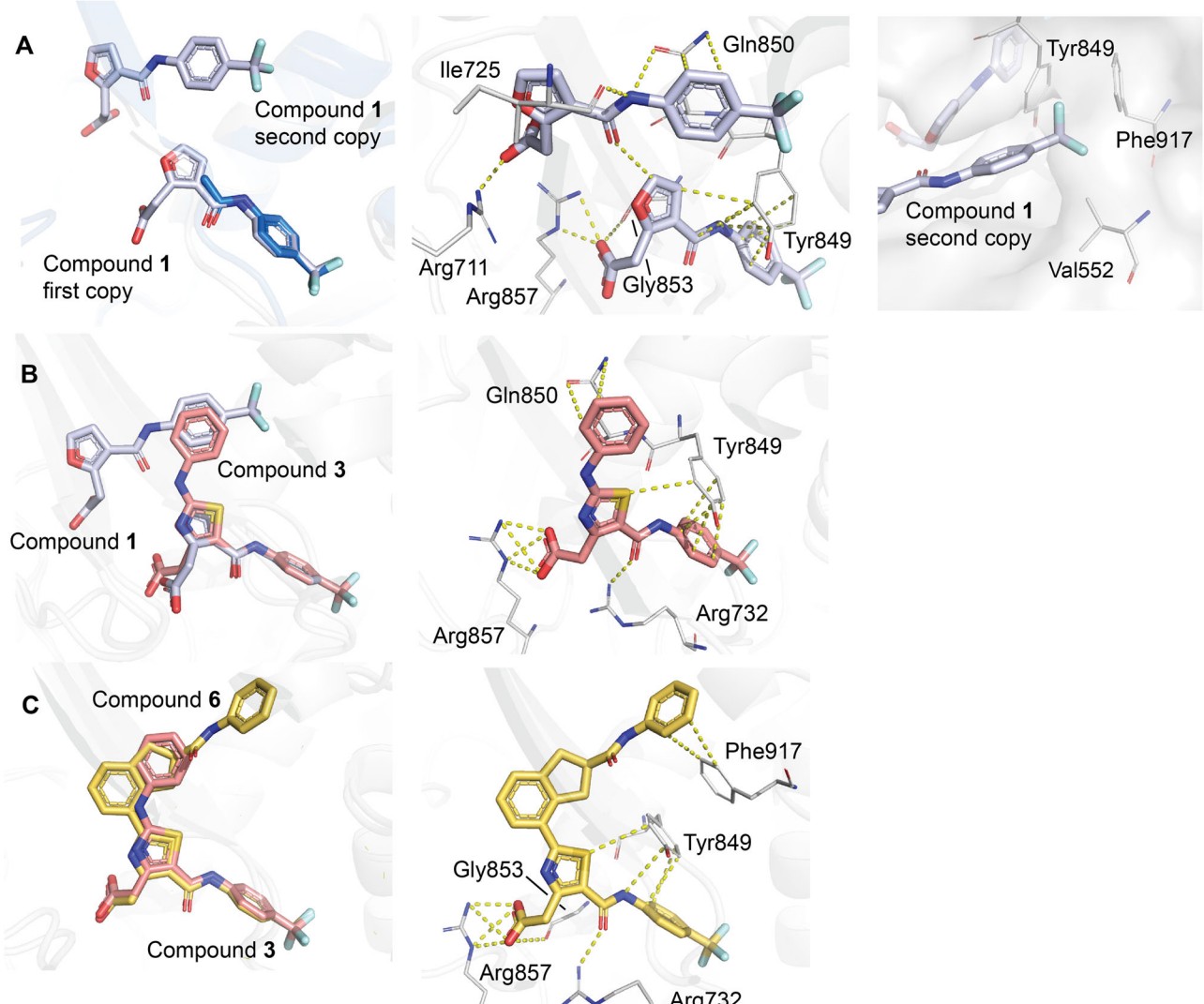

**Fig. 7 | Binding sites of Compounds 1, 3, and 6. A** Two copies of **compound 1** (pale blue) bound to WRN Form D, with one copying overlaying with **fragment 1**. Interactions between the two **compound 1** copies and WRN are shown in yellow dashes. Residues Val552, Tyr849, and Phe917 are nearby **compound 1** trifluoromethyl. **B Compound 3** (salmon) is shown overlaid with **compound 1** (pale blue). Interactions between **compound 3** and WRN are shown in yellow dashes. **C Compound 6** (yellow) overlaid with **compound 3** (salmon). Interactions between **compound 6** and WRN are shown in yellow dashes.

illustrated in the overlay with the WRN-AMP-PNP structure (Fig. 8B). Despite the significant change in overall conformation, the binding pocket itself in Form E shares 16 of the 18 residues of that in the biologically-validated Form B. **Compound 5** binds with the trifluoro moiety located in the anchoring fragment pocket, interacting with Thr728, Phe730, Arg732, and Arg857 (Fig. 8C). The electron density corresponding to the terminal pyrimidine is weak (Fig. 4), suggesting that this ring may adopt two different orientations, rotating around the bond between the carbonyl carbon atom and the piperazine nitrogen atom; it is modeled here in both conformations. As **compound 5** forms a hydrogen bond with Arg 857 which also interacts with AMP-PNP, we hypothesize that ATP and **compound 5** cannot bind WRN simultaneously.

The transition of the WRN domains from the ATP-bound Form A to Form B/D involves an approximate 180° rotation around the linker (Fig. 9A). In Form E, determined by co-crystallization of WRN with **compound 5**, the configuration of D2 relative to D1 also results in an approximate 180° rotation from Form A, but this rotation occurs along a different axis than that from Form A to Form D (Fig. 9B). The conformations of the domains in Forms D and

E reveal an approximate 90° rotation between these two forms (Fig. 9C).

## Discussion

The path to identifying bona fide small molecule inhibitors of helicases has proven challenging, as the most common high-throughput screening approaches used to date have relied on biochemical enzymatic assays plagued by high false positive hit rates, driven at least in part by the susceptibility of these assays to non-specific or nuisance activities, common with unoptimized small molecule screening positives. Identifying and triaging artifacts rather than actual hits is a common challenge, as nuisance compounds can induce a plethora of non-technology-related interferences and nonspecifically inhibit interrogated assay readouts[35,36]. Recently, Hauser et al. have shown that compounds such as ML216, NSC19630, and NSC617145 manifest their activity not by specific interaction with WRN but by nonspecific inhibition due to the presence of impurities in the test samples (ML216) or through covalent labeling (NSC compounds)[22]. In further support, we observe these challenges within biochemical characterization presented here, where steep hill slopes, symptomatic of poor

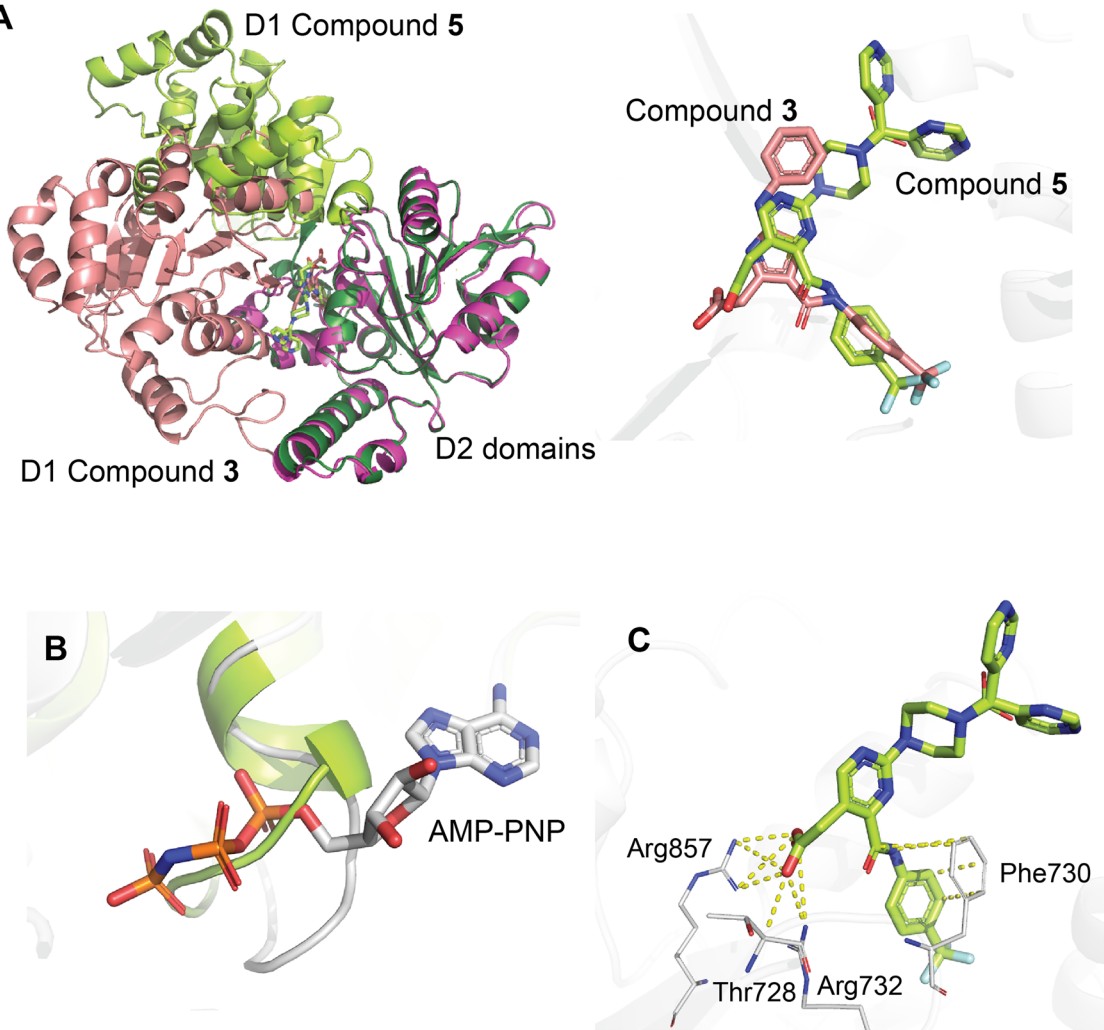

**Fig. 8 | The discovery of WRN Form E in the presence of Compound 5.**
**A** Compound **5** (green), designed from **compound 3** (salmon), binds to an additional form of WRN – Form E. The two ATPase domains are rotated by approximately 90° between Form D and Form E. When overlaid on the D2 domain, compound **5** overlays very closely to **compound 3**. **B** A loop consisting of residues Val570-Ser578 that forms the ATP binding site in Form A (white) is displaced in the presence of **compound 5**, closing off the pocket. **C** Interactions between **compound 5** and WRN are shown in yellow dashes.

compound property-driven inhibition, were observed in BLM ATPase quantitation for **compound 2, compound 6** and **compound 7**. Hence, we hypothesized that shifting the screening strategy for WRN to biophysics-based approaches would identify non-covalent fragment allosteric hits that could be validated as WRN inhibitors with multiple parallel methods to limit the number of artifacts acting through nuisance mechanisms[37]. In our FBLD workflows, we leveraged parallel bioNMR- and SPR-based screens to enable the discovery of low-affinity but specific hit molecules in two allosteric sites on WRN. We were successful in developing chemical matter in the binding site of fragment 1 into functional inhibitors against this challenging drug target. High-resolution X-ray crystal structures expedited the fragment design process by offering insights into protein-ligand interactions, confirming the recently described complex dynamics of WRN, and enabling us to visualize an additional ligand-bound WRN conformation that may be useful for future inhibitor design campaigns.

While helicases are recognized for their considerable flexibility, our work further supports the dynamic nature of WRN by revealing two conformations, Forms D and E. Despite these significant conformational variations, the overall RMSDs of the Cα backbone atoms for the D1 and D2 domains comparing AMP-PNP-bound (Form A), **fragment 1**-bound (Form D), and **compound 5**-bound (Form E)

structures are: approximately 0.7 Å (0.68 Å for AMP-PNP:**fragment 1** over 167 atoms; and 0.63 Å for AMP-PNP:**compound 5** over 164 atoms) and about 0.5 Å (0.47 Å AMP-PNP:**fragment 1** over 127 atoms; 0.53 Å AMP-PNP:**compound 5** over 129 atoms), respectively[38]. Previous studies of RecQ helicases have described the flexible aromatic loop linking the HDRC domain to the WH motif[39]; however, our findings emphasize the substantial flexibility of the linker region between D1 and D2 domains (residues Thr728-Asn734), allowing for the adoption of diverse conformations while maintaining structural consistency within each ATPase domain (Fig. 10A). The orientation of the C-terminal helix, as highlighted in Fig. 10B, further underscores the wide range of WRN conformations that are currently known[7,24,25,39].

In addition to the two allosteric pockets in Form D and Form E, we have also discovered an additional allosteric pocket in the **fragment 3**-bound structure, situated approximately 21 Å from the Form D allosteric pocket. SPR, NMR, and structural studies have validated the direct interactions between **fragment 3** and WRN; however, developing an inhibitor in this pocket poses a challenge, as **fragment 3** is currently inactive in our biochemical assays. Furthermore, the structure featuring **fragment 4** shows a clear density region for fragment 4, along with another density region that we modeled as a ligand byproduct, within the remnants of the ATP pocket. This highlights the capabilities of

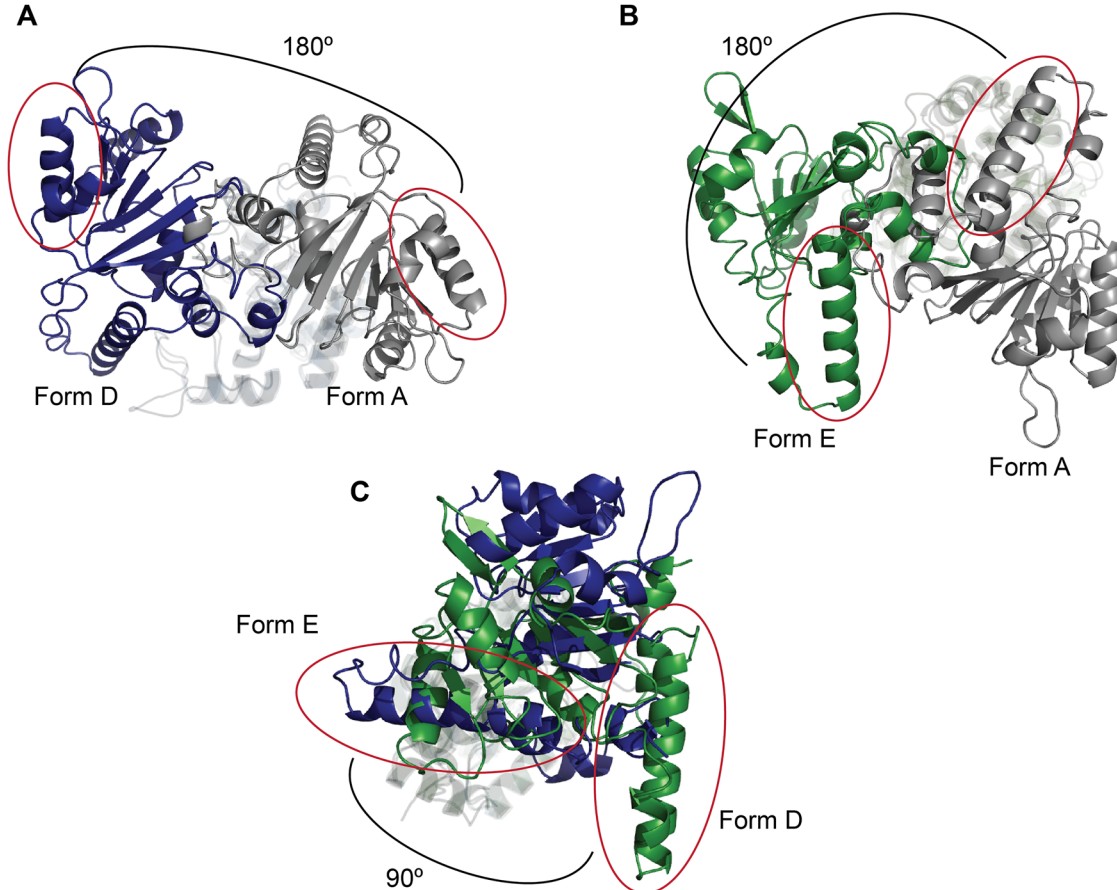

**Fig. 9 | Comparison of WRN conformations Form A, D, and E.** The three forms highlighted here aligned on D1 domains, with Form A in gray, Form D in blue, and Form E in green. The differences in domain configurations can be thought of as rotations around the linker residues connecting D1 and D2. Each structure has a red oval covering the same C-terminal helix containing residues Phe888-His904 to help orient and visualize these rotations. **A** An approximate 180° rotation of D2 domains is seen between Form A and Form D. **B** There is also an approximate 180° rotation of the D2 domains between Form A and Form E (green). However, the rotation axis is different than that seen between Form A and Form D. **C** Form D and Form E structures differ by a rotation of approximately 90° around the linker.

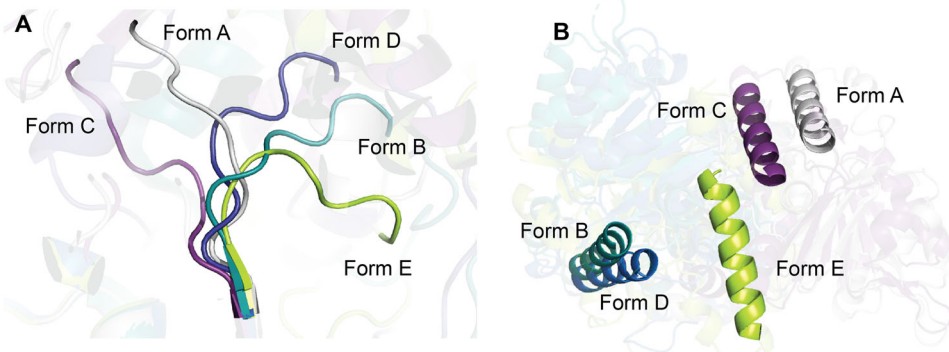

**Fig. 10 | Overlays of all known WRN conformations.** Representatives of all known forms of WRN aligned on their D1 domains. Form A in gray, Form B in purple, Form C in mint, Form D in blue, and Form E in lime green. **A** Linker residues Cys727-Asn734 show different trajectories of connection between the D1 and D2 domains of the various WRN forms. **B** The same C-terminal helix as in Fig. 9 highlights overall conformational differences.

FBLD workflows in identifying novel ligands of diverse shapes and charges within both established orthosteric pockets and allosteric sites.

While the Form D fragment-bound structures described in this work exhibit three-dimensional similarities to Form B (PDB 8PFO)[24], the overall conformation of WRN is distinct. The RMSDs within the binding pockets and surrounding protein residues are 0.31 Å (174 atoms) for the D1 domains and 0.38 Å (142 atoms) for the D2 domains, indicating that there are no significant conformational differences in the main chain or side chains within each individual domain. However, when superimposing the D1 domains of Form D and Form B it is evident that the D2 domains are offset from each other by approximately

9 Å at their most distal point (Fig. 5C). It is important to note that the protein constructs and chemical matter are different between the two structures, as are the space groups and unit cells (data for Form D found in Methods Data collections and refinement statistics table). These differences likely play a role in the similar, yet distinct, orientations of the ATPase domains, underscoring WRN's flexibility and its anticipated ability to adopt various conformations in vivo while performing its enzymatic functions.

Unexpectedly, the binding of structurally similar analogs, **compounds 5** and **6**, resulted in yet another distinctive conformation of WRN when **compound 5** is bound, even though they bind to the same anchoring region of the allosteric pocket and share over 85% of the pocket residues in both forms. Since crystal structures provide snapshots of protein conformations and arrangements, it is reasonable to conclude that these structures capture only a few states of WRN as it translocates along the DNA and conducts its helicase and exonuclease functions. Consequently, modeling potential intermediate states or predicting the accessibility of binding pockets during these transitions presents a significant challenge. We have provided Supplementary movie 1 showing interpolated transitions of WRN between the three forms, Form A, Form D, and Form E. Alignments of the ATPase D1 and D2 domains of **fragment 1** and **compound 5** with the structure of BLM helicase with bound DNA (PDB 4CGZ)[40] indicate that DNA is likely able to bind to the two additional conformations of WRN, Form D and Form E, respectively (Figure SI 10). Consequently, the disruption of the ATP binding pocket or the prevention of domain rearrangement to achieve an ATP-binding competent conformation are both plausible mechanisms of inhibition.

The structures reported here provide further insights into the mechanism of WRN helicase activity, which is believed to be unique among the RecQ helicase family. Biochemical data also suggest that WRN can adopt unique conformations during DNA unwinding. For instance, WRN's unwinding of a forked substrate with a 25 bp double-stranded DNA (dsDNA) region was found to necessitate higher ATP concentrations compared to those required for direct ATPase activity measurement. Conversely, when using forked substrates with a shorter 10 bp dsDNA region, the ATP concentrations associated with either DNA unwinding or ATPase activity are comparable. These findings indicate either a potential role for pausing or a non-productive ATP hydrolysis mechanism in the unwinding of dsDNA by WRN, or are a consequence of the enzyme's apparent strand annealing activity, a functional property shared by many of its structurally analogous DEAD-Box RNA helicases[19]. Moreover, single-molecule studies support the notion that while WRN is more adept at repetitively unwinding short duplex DNA regions, the helicase core pauses and reciprocates back along the DNA when unwinding a longer, 29 bp substrate[41–43]. In contrast, the homologous RecQ family member BLM helicase exhibits similar ATP concentration dependences for both ATP hydrolysis and the unwinding of forked substrates containing 25 bp dsDNA regions[41–43]. BLM has been observed to employ a strand-switching mechanism instead of a pausing/reciprocating unwinding mechanism seen in WRN[44]. Therefore, supported by our most potent WRN ATPase inhibitor, **compound 4** and its ≥ 100-fold selectivity for WRN over BLM (Table S2), we hypothesize that WRN may adopt unique conformational states that facilitate pausing and reciprocation during unwinding, which could be captured and stabilized in an inactive state by small molecule inhibitors. Further in vitro studies directly probing helicase unwinding will be necessary to elucidate additional details and enhance our understanding of the unique characteristics of WRN function.

In summary, the work presented here demonstrates the power of FBLD in providing robust starting points for highly challenging targets where high-throughput screening approaches rooted in biochemical enzymatic assays had a low success rate in delivering validated hits. The unique ability of fragments to identify multiple binding sites on target proteins was utilized for this project to discover two allosteric sites in varying conformations of WRN. To prevent the elaboration of silent binders, we leveraged knowledge from X-ray data for ADP-bound WRN in form A and prioritized work on **fragments 1 and 2** that bound to an allosteric site formed by breaking apart the ATP site in form D and engaging Arg857, which makes key interactions with the triphosphate group of ATP. This approach has proven to be highly fruitful, as parallel to our work, a clinical candidate HRO761 was discovered that bound to the same allosteric binding pocket on WRN while acting through mixed ATP competition via allosteric binding as expected. Interestingly, the substituted N-[4-(trifluoromethyl)phenyl]acetamide scaffold that we discovered in our bioNMR screen (fragment 1) is part of HRO761 that anchors it into the allosteric binding site and was optimized, independently of our work. Moreover, it has been shown that HRO761 can selectively inhibit the growth of MSI cancer cells in vitro as well as in vivo, proving the biological relevance of the chemical matter binding to this allosteric binding site.

Our findings underscore how structural studies, when combined with FBLD, offer valuable insights into the conformational changes and ligand binding complexities of targets like WRN helicase, which exhibit multiple conformations and activation states. We anticipate a growing interest in the exploration of synthetic lethal targets, with the inhibition of WRN remaining a significant focus within the oncology field. As a study that details the exploration and modification of fragments aimed at discovering novel and potent WRN inhibitors, the work provides value in advancing small-molecule drug discovery for WRN, and potentially other hard-to-target helicases, through this alternative methodology.

## Methods

### Gene expression and protein purification

The helicase domain of Human Werner-Syndrome (WRN) protein UniProt ID Q14191 for SPR, X-ray, biochemistry, and bioNMR was expressed using the residues and tags as described in the Supporting Table SI 1.

For SPR measurements, the N-terminal GST-PRE and C-terminal AVI-tagged human WRN (500–946) was cloned into pFastBac1. Bacmids from this vector were transfected into Sf9 insect cells to generate baculovirus. Protein from baculovirus was expressed in Sf21 cells in Sf-900™ II SFM (ThermoFisher Scientific−Cat #11497013) + 10% FCS at 26 °C for 64 h in a 5 L ReadyToProcess WAVE™ 25 bioreactor system. Cells were pelleted down and stored at −80 °C for future use. All purifications were carried out on ice or at 4 °C unless stated.

For purification, frozen cells were thawed and resuspended in buffer A: 50 mL of 2x PBS pH 7.4, 500 mM NaCl, 10% Glycerol, 5 mM DTT, 0.1% CHAPS, supplemented with 1 Protease Inhibitor Tablet (Roche) and 20 U/mL Benzonase. Cells were lysed and homogenized prior to a centrifugation step for 40 min and 75,000 x g. The lysate was loaded onto a 15 mL Glutathione Sepharose column in batch mode and washed with 4 CV of buffer A. WRN protein was eluted with a step increase of GSH concentration to 25 mM, followed by 4 CV wash with buffer B; 50 mL of 2x PBS pH 7.4, 500 mM NaCl, 10% Glycerol, 5 mM DTT, 0.1% CHAPS, 25 mM GSH. Fractions with the highest content and purity of WRN protein were pooled and dialyzed overnight using Spectra/Por 1 Dialysis Tubing 6–8 kDa against buffer C (10 mM HEPES-NaOH, pH 7.5, 50 mM NaCl, 1 mM DTT, 5% glycerol, 0.1% CHAPS) in the presence of GST-HRV3C protease for tag cleavage. Cleaved protein sample from the initial purification step was reloaded onto a 5 mL Heparin-column (2 × 5 mL and washed with buffer C containing 150 mM NaCl. WRN was eluted in a 20 CV gradient to buffer D (10 mM HEPES/NaOH, pH 7.5, 1 M NaCl, 1 mM DTT, 5% glycerol, 0.1% CHAPS). Collected protein fractions were concentrated to 12.4 mg/mL in an Amicon ultra concentrator (50 kDa cut-off) and loaded at 1 mL/min onto a HiLoad 26/60 Superdex 200 column pre-equilibrated in buffer E (10 mM HEPES-NaOH pH 7.5, 500 mM NaCl, 5% Glycerol, 1 mM TCEP).

The sizing exclusion chromatogram (SEC) showed one main peak, which corresponded to monomeric, cleaved WRN protein. Suitable fractions were pooled and concentrated as above to a final concentration of 5.2 mg/mL (Figure SI 2B). Aliquots were snap frozen in liquid nitrogen and stored at −80 °C.

The expression and purification workflow described above for the WRN construct encompassing residues 500–946 delivered high-quality material for SPR studies; however, further adjustment of the construct and workflow was necessary to support bioNMR and X-ray studies. Hence, for X-ray, biochemistry, and bioNMR application,s the His-FLAG tag followed by a Thrombin cleavage site was fused to the N-terminus of the helicase domain and cloned into the pFastBac1 vector for baculovirus production and large-scale expression performed in Sf21 insect cells in Sf-900 II SFM medium + 10% FCS at 26 °C for 64 h in a 5 L ReadyToProcess WAVE™ 25 bioreactor system. Cells were harvested by centrifugation and stored at −80 °C for future use. All purifications were carried out on ice or at 4 °C unless stated.

For purification, frozen cells were removed from −80 °C storage, thawed and resuspended in lysis buffer F (50 mL 2x PBS pH 7.5, 500 mM NaCl, 1 mM TCEP, 10% glycerol, 40 mM imidazole, 0.1% CHAPS) supplemented with DNase I and Protease Inhibitor Tablet (Roche). Cells were lysed and homogenized prior to a centrifugation step for 40 min and 75,000 x g. The cleared lysate was mixed with 20 mL Ni-Sepharose beads and incubated for 1.5 h while stirring. Beads were washed with 5 column volumes of buffer F and filled into an XK column. The XK column was washed with lysis buffer containing 50 mM imidazole and eluted with a 10 CV gradient to buffer G (2x PBS pH 7.5, 500 mM NaCl, 1 mM TCEP, 10% glycerol, 500 mM imidazole, 0.1% CHAPS).

WRN eluted in imidazole gradient was incubated with 10U thrombin/mg protein in dialysis in buffer E overnight to remove His-FLAG tag. The dialyzed protein was passed through a Ni-Sepharose column again to remove residual fusion protein. Flow-through was concentrated, diluted 6-fold with heparin buffer C (10 mM HEPES-NaOH, pH 7.5, 50 mM NaCl, 1 mM TCEP, 5% glycerol, 0.1% CHAPS), applied on heparin column (2 × 5 mL) and washed with heparin buffer C containing 100 mM NaCl. WRN was eluted in a 20 CV gradient to buffer D (10 mM HEPES-NaOH, pH 7.5, 1 M NaCl, 1 mM DTT, 5% glycerol, 0.1% CHAPS). Elution fractions were pooled, concentrated to 31.1 mg/mL, and loaded on HiLoad 26/60 Superdex 200 column equilibrated in SEC buffer E (10 mM HEPES-NaOH, pH 7.5, 500 mM NaCl, 5% Glycerol, 1 mM TCEP). Elution fractions were analyzed through Tricine-SDS-PAGE, and relevant fractions were pooled, concentrated, flash-frozen in liquid nitrogen and stored at −80 °C (Figure SI 2A).

Human RecQ-like DNA helicase (BLM), Uniprot ID P54132, protein was designed as an N-terminally truncated variant (642-1296) and modified by fusing a 6xHis-Tev tag to the N-terminus of the selected BLM boundary (MGHHHHHHHENLYFQG). The sequence was cloned into pET28b vector for expression in BL21 (DE3) cells obtained from NEB). Transformed BL21 (DE3) cells were grown in LB medium (Teknova) in the presence of 50 μg/mL Kanamycin and 150 μM ZnCl₂. Expression of BLM was initiated by induction of the cells with 0.3 mM IPTG at an OD600 between 0.5-0.6. Cells were grown for 18 h at 37 °C, shaking at 200 rpm in a Multitron shaker (Infors HT) and harvested via centrifugation at 4000 x g for 20 min at 4 °C. Pellets were frozen and stored at −80 °C.

A cell pellet representing an expression volume of 10 L was defrosted, and a weight of app. 50 g cell pellet was resuspended in 250 mL lysis buffer (50 mM Tris pH 7.5, 500 mM NaCl, 10% glycerol, 1 mM TCEP, 0.1% Triton X-100, 20 mM imidazole, 10 U/mL Benzonase and 5 protease inhibitor cocktail tablets (Roche)) and sonicated for 150 s in 5 s on/30 s off intervals with an amplitude of 40%. The lysate was cleared by a centrifugation step (230,000 x g for 45 min at 4 °C). 250 mL of cleared lysate was mixed with 10 mL HisPur resin and

incubated in batch for 2 h at 4 °C while stirring before being packed into a column. The column was washed with 4 CV at 4 mL /min in buffer C (50 mM Tris pH 7.5, 500 mM NaCl, 10% glycerol, 1 mM TCEP, 0.1% Triton X-100, 20 mM imidazole, protease inhibitor tablets at 1/50 mL). The following gradient steps were applied: 6 CV with 5% buffer D (buffer C with an imidazole concentration increased to 250 mM), 15 CV gradient from 0% B to 100% B, followed by 3 CV of 100% buffer B. 2 mL fractions were collected with a final volume of 122 mL at 1.24 mg/mL. The pooled fractions were diluted with 50 mM Tris pH 7.5, 5% glycerol to a final NaCl concentration of 250 mM and loaded onto a 5 mL Hi Trap Heparin HP column at 4 mL/min. The column was washed with 5 CV buffer C (20 mM Tris, pH 7.5 5% glycerol, 250 mM NaCl). Protein elution was conducted using a 20 CV gradient from 0%-100% buffer D (20 mM Tris pH7.5, 5% glycerol, 800 mM NaCl). Followed by 5 CV of 100% buffer D. 30 mL fraction volume at 1.1 mg/mL was pooled and split equally into two sub-pools (Pool 1 and 2).

Pool 1 (approximately 25 mg protein in 15 mL) was loaded onto a Hi Load 26/60 Superdex 200 column pre-equilibrated in 20 mM Tris, pH 7.5, 5% glycerol, 500 mM NaCl. 18 mL total volume was collected at 0.72 mg/mL and concentrated to approximately 6 mL in an Amicon ultra concentrator (50 kDa cut-off) with a final concentration of 2.2 mg/mL. 50 μL aliquots were frozen and stored at −80 °C. Pool 2 (approximately 25.5 mg protein) was mixed with 1.28 mg His-tagged Tev protease in approximately 15 mL volume and incubated for 17 h at 4 °C. Cleavage was confirmed by SDS-PAGE and anti-His Western Blot, using anti-His6-Peroxidase from mouse IgG1 (Roche 11965085001) in a 1:500 dilution, before being further processed. The cleaved sample was loaded onto a Hi Load 26/60 Superdex 200 column pre-equilibrated in 20 mM Tris pH 7.5, 5% glycerol, 500 mM NaCl. 18 mL total volume was collected at 0.72 mg/mL and concentrated to approximately 4.5 mL in an Amicon ultra concentrator (50 kDa cut-off) with a final concentration of 2.7 mg/mL. 50 μL aliquots were frozen and stored at −80 °C.

## BioNMR ¹⁹F fragment screening

The 1020 fragments from internal fluorine-containing collection were selected based on their measured ¹⁹F chemical shifts, purity >90%, solubility >250 μM, and lack of aggregation at 50 μM in the CPMG experiments. Fragments were pooled in the mixtures of up to twenty-one compounds covering a range of ¹⁹F chemical shifts, ensuring at least 1 ppm separation between mixture components. The control samples were prepared at 50 μM of each mixture component, with DMSO-D6 concentration always matched to 2.5% final volume. Samples to evaluate binding were prepared in 25 mM HEPES-NaOH, 50 mM NaCl, 2 mM MgCl₂, pH 7.4 buffer with protein concentration at 10 μM and each mixture component at 50 μM, with DMSO-D6 concentration matched to 2.5% final volume in each sample. The ¹⁹F CPMG relaxation experiments were recorded at 295 K on a Bruker AVANCE III 600 MHz NMR spectrometer equipped with a 5 mm QCI-F cryoprobe. Bruker CPMG T2 pulse program (19Fcpmg_screen) was implemented employing ca-WURST-20 adiabatic refocusing pulse, 20 ms and 160 ms total relaxation time, with 112 scans and proton decoupling during acquisition. The NMR spectra were analyzed using the Fragment-Based Screening (FBS) module in Topspin 3.5pl7. Hits were identified by signal intensity changes and selected for singleton confirmation when the quotient $Q_{bind}$ was less than 0.5[45].

$$Q_{bind} = \frac{Intensity\ Ratio^{+target}}{Intenisty\ Ratio^{-target}} \quad (1)$$

where

$$IntensityRatio = \frac{Peak\ Integral^{CPMG\ at160ms}}{Peak\ Integral^{CPMG\ at20ms}} \quad (2)$$

Because each fragment in the mixture had a known unique chemical shift, the hits could be simply identified by matching the chemical shift of the hit to that defined in the mixture reference spectra.

Each hit identified in the mixture was further submitted to a set of NMR experiments in single-molecule mode. First, the $^{19}$F CPMG experiments were performed for all hits in isolation under conditions identical to the screening setup. Next, confirmed binders were subjected to a proton-based suite of experiments to gain confidence in the binding: $^{1}$H $T_2$ CPMG, STD, and waterLOGSY. All three experiments were acquired on a Bruker AVANCE III 600 MHz NMR spectrometer equipped with a 5 mm QCI-F cryoprobe with the same experimental conditions, where for compound-bound samples, the WRN protein was reconstituted at 10 μM in the 25 mM HEPES-NaOH, 50 mM NaCl, 2 mM MgCl$_2$, pH 7.4 buffer. Compounds were added at a final concentration of 200 μM, and DMSO-D6 was always matched to 2.5% final sample volume. In compound QC samples, fragments were reconstituted at 200 μM in the 25 mM HEPES-NaOH, 50 mM NaCl, 2 mM MgCl$_2$, pH 7.4 buffer with 2.5% DMSO-D6. The STD (stddiffesgp.3) experiments were performed with a train of 40 Gaussian bell-shaped selective pulses of 50 ms length, and 2 s recycle delay; the on- and off-resonance frequency for selective irradiation of protein were set to 0 ppm and −40 ppm, respectively. The number of transients was set to 256. Data were automatically processed using the proc_std Bruker AU program. The $^{1}$H $T_2$ CPMG (cpmg_esgp2d) experiments were implemented with pulse sequence cpmg_esgp2d and CPMG filters set to 20 and 400 ms. The number of scans was 128. In water-LOGSY (ephogsygpno.2) experiments were acquired using the pulse sequence as described by Dalvit et al.[46,47]. In summary, experiment parameters were: 256 number of transients, water excitation performed using a 180 degree selective Gaus1_180r.1000 pulse followed by a 2 s mixing time and 5 s relaxation delay. Water suppression was achieved with a 2-ms 180-degree selective Sinc1.1000 pulse at the H2O frequency. Protein resonances were suppressed using CLEANEX spinlock with mixing time set to 30 ms. For STD, waterLOGSY, and $^{1}$H $T_2$ CPMG experiments, the peak intestines retrieved from spectra in the presence of protein were compared with the reference spectra for the individual compounds in the buffer. Compounds were considered binders when the STD double difference was more than 10% of the reference signal, water-LOGSY gave a positive signal, and $Q_{bind}$ was less than 0.5 in the $^{1}$H $T_2$ CPMG experiment.

## SPR binding assays

The diversity set of 500 fragments from our internal non-fluorine containing collection was subjected to a series of SPR binding assays. The SPR binding assays were carried out using a high-throughput Biacore 8 K + SPR system (Cytiva29283382), with biotinylated WRN protein immobilized to the biosensor chip surface using a proprietary biotin capture method (Biotin Capture Kit; Cytiva28920234). All reagents were kept at 8 °C in the reagent compartment. Briefly, the kit Series S CAP Sensor chip was docked to the instrument and prepared according to the manufacturer's protocol. A solution of 100 μg/mL WRN (diluted in 10 mM HEPES-NaOH pH 7.4, 500 mM NaCl, 5 mM MgCl$_2$, 1 mM TCEP, 0.01% Tween20) was injected into flow cell 2 (all 8 channels) for 2 min at a flow rate of 10 uL/min to achieve an immobilization density of ~3000 RU. Flow cell 1 was reserved as the reference cell used for reference cell correction. The sensor chip was subsequently equilibrated at 20 °C in running buffer (10 mM HEPES-NaOH, pH 7.4, 150 mM NaCl, 5 mM MgCl$_2$, 1 mM TCEP, 0.01% Tween 20, 1% DMSO) prior to analysis of test compounds. For the primary single-point screening of the non-fluorinated fragment collection, compound stocks of 100 mM in 100% DMSO were diluted 100-fold in assay buffer to a 1 mM final test concentration with 1% DMSO in the final 100 ul sample volume. This was carried out using an automated liquid handler (Echo 655). For the 4-point concentration response test,

multiple concentrations were also prepared from the 100 mM compound stocks using the Echo 655, generating 4 concentration points for each fragment samples including 125 μM, 250 μM, 500 μM and 1000 μM. This was performed with the appropriate back-fill to reach a final 1% DMSO in the final 100 μl sample volume. Each test fragment was injected for 30 s at a 40 μL/min flow rate (association phase). This was followed by a 45 s dissociation phase using running buffer. Each fragment was tested at 1 mM concentration in two independent trials ($N = 2$). To account for refractive index mismatch due to slight differences in the amount of DMSO in each test fragment injection, a DMSO solvent correction was applied to all data sets. DMSO standard curve was prepared using 5 concentrations (0.5% increments from 0% DMSO to 2% DMSO.

Binding occupancy for each test fragment was calculated using the ratio of observed binding RU (at the end of each test fragment injection) and the theoretical maximum occupancy based on a 1:1 binding model. The primary screen hits were then prioritized based on binding occupancy and their binding kinetics profile. Hits with less than 20% occupancy were deprioritized. The remaining hits were clustered according to binding kinetic profile. Each cluster was ranked (Fig. 2) as follows: a) *Rank 1*, fast-on, fast-off binding kinetics; b) *Rank 2*, biphasic binding profile with predominantly fast-on, fast-off binding kinetics and a minor component (roughly ≤ 20% of the binding profile) of slow-on, slow-off binding kinetics; c) *Rank 3*, Significant (> 20% of the binding profile) slow kinetic profiles; d) *Rank 4*, super-stoichiometric/non-saturable binding. Fragments in ranks 3 and 4 were deprioritized. Prioritized hits (Rank 1 and 2) were then validated in Tier 2 dose-response binding analysis (4 concentrations, two-fold dilution series with top concentration at 1 mM).

In the follow-up studies of the chemistry-derived chemical matter, the SPR binding assays were carried out using a high-throughput Biacore 8 K + SPR system (Cytiva), with biotinylated WRN protein immobilized to the biosensor chip surface using a proprietary biotin capture method (Biotin Capture Kit; Cytiva). All reagents were kept at 8 °C in the reagent compartment. Briefly, the kit Series S CAP Sensor chip was docked to the instrument and prepared according to the manufacturer's protocol. A solution of 25 μg/mL WRN (diluted in 10 mM HEPES-NaOH pH 7.4, 500 mM NaCl, 5 mM MgCl$_2$, 1 mM TCEP, 0.01% Tween20) was injected into flow cell 2 (all 8 channels) for 5 min at a flow rate of 10 uL/min to achieve an immobilization density of ~2300 RU. Flow cell 1 was reserved as the reference cell used for reference cell correction. The sensor chip was subsequently equilibrated at 20 °C in running buffer (10 mM HEPES-NaOH, pH 7.4, 250 mM NaCl, 5 mM MgCl$_2$, 1 mM TCEP, 0.01% Tween 20, 1% DMSO) prior to analysis of test compounds. Compound stocks of 10 mM in 100% DMSO were diluted 400-fold in assay buffer to a 25 μM final test concentration, while stocks of 50 mM in 100% DMSO were diluted 100-fold in assay buffer to a 500 μM final test concentration with 1% DMSO in the final 100 μl sample volume. This was carried out using an automated liquid handler (Echo 655). For the 4-point concentration response, multiple concentrations were prepared using the Echo 655, generating 4 concentration points for each compound, including 25 μM, 8.25 μM, 2.75 μM, 1.0 μM, 0.25 μM, 0.125 μM, 0.031 μM, and 0.016 μM, or 500 μM, 250 μM, 125 μM and 62.5 μM. This was performed with the appropriate back-fill to reach a final 1% DMSO in the final 100 μl sample volume. Each test compound was injected for 120 s at a 20 μL/min flow rate (association phase). This was followed by a 300 s dissociation phase using running buffer. To account for refractive index mismatch due to slight differences in the amount of DMSO in each test injection, a DMSO solvent correction was applied to all data sets. DMSO standard curve was prepared using 5 concentrations (0.5% increments from 0% DMSO to 2% DMSO.

All data were analyzed using Biacore Insight evaluation software, version 5.0.18.22102. The sensorgrams were fit with a 1:1 steady-state affinity binding model to obtain $K_d$, when applicable.

## ATPase assay

To measure WRN ATPase activity, the accumulation of ADP was quantitated via ADP-Glo Assay kit (Promega) in all black NBS low-volume 384-well microplates (Corning, 3820). For the construction of assay-ready plates, 200 nL of each compound was dispensed in DMSO via three-fold serial dilution (100 μM, the ~~Top~~ highest concentration) into assay plates using an Echo acoustic dispenser. All assay reaction solutions were prepared in 1X assay buffer (25 mM HEPES-NaOH, pH 7.3, 50 mM NaCl, 2 mM $MgCl_2$, 0.01% Tween-20, 0.5 mM TCEP, 1% DMSO). All WRN compound inhibition reactions were assembled in a final volume of 20 μL by preincubating (250 pM, final) WRN helicase core (500–942) with the respective compound (Figure SI 4), for 30 min in the presence of (100 nM, final) single stranded DNA (IDT; 5'-TTTTTTTTTTTTTT TTTTTTTTTTTTTTTTCGTACCCGATGTGTTCGTTC-'3), which is required for ATP hydrolysis, prior to reaction initiation by the 1:1 addition of 1X $K_m$ ATP (50 μM, final). All BLM compound inhibition reactions were assembled in a final volume of 20 μL by preincubating (500 pM, final) BLM helicase core (642–1296) with the respective compound (Figure SI 4), for 30 min in the presence of (30 nM, final) single-stranded DNA and initiated by the 1:1 addition of 1X Km ATP (50 μM, final). Both BLM and WRN ATPase reactions were run for 1 h prior to quenching by the addition of the ADP-Glo kit reagents. High luminescence (enzyme + substrates + DMSO) represents the signal from max ATP hydrolysis (no ATPase inhibition), while low luminescence (substrates + DMSO or WRN + Substrates + 20 mM positive inhibition control) represents no ATPase activity/complete inhibition. Compound concentration response curves were fit by the 4 P (Spotfire) inhibition model to extract $IC_{50}$, Hill slope, and % maximum inhibition. Reported errors represent the nonlinear least squares 95% confidence interval of the extracted parameter.

## Ligand docking

Crystal structures were prepared using the Protein Preparation Wizard panel in Maestro 13.1 (Schrodinger Suite 2022-1). Grids of these structures were generated using the receptor grid generation panel of Maestro 13.1. The centroid of co-crystallized ligands was used to define the binding site. Ligands were prepared using the LigPrep panel of Maestro 13.1. All possible stereoisomers were generated. All docking calculations were performed using Glide (Schrodinger Suite 2022-1) using standard precision and flexible ligand sampling.

## Crystal growth and preparation

**AMP-PNP structure.** Co-crystals of hWRN were grown in the presence of AMP-PNP by sitting drop vapor diffusion at 4 °C. Prior to drop set-up, hWRN was incubated with 2 mM AMP-PNP for 2 h on ice. The drops were set in a 1:1 ratio (0.1 μL: 0.1 μL) of protein mix (10 mg/mL in 1 mM TCEP, 5% (v/v) glycerol, 10 mM HEPES-NaOH pH 7.5, and 500 mM NaCl) to mother liquor containing 30% Jeffamine ED-2003, 0.2 M NaCl, 0.2 M NDSB-201, and 0.1 M MES/NaOH pH 6.0. Crystals were obtained after 3 weeks. Crystals were flash-vitrified directly from the drop and measured at a temperature of 95 K.

## Fragment 1

Co-crystals of hWRN were grown in the presence of ligand via hanging drop vapor diffusion at 4 °C. Prior to drop set-up, hWRN was incubated with 20 mM **fragment 1** for 16 h on ice. The drops were set in a 1:1 ratio (0.5 μL: 0.5 μL) of protein mix (10 mg/mL in 1 mM TCEP, 5% (v/v) glycerol, 10 mM HEPES-NaOH pH 7.5, and 500 mM NaCl) to mother liquor containing 20% (w/v) PEG 6000, 0.1 M HEPES-HCl pH 6.25-6.5. Crystals were obtained after 1 week. Crystals were flash-vitrified in a solution of mother liquor + ligand with 20% (v/v) ethylene glycol and measured at a temperature of 95 K.

## Apo Form D, fragment 2, fragment 3, fragment 4, compound 1, compound 3, and compound 6

Apo crystals in Form D were grown by hanging drop vapor diffusion at 4 °C. The drops were set in a 1:1 ratio (0.5 μL: 0.5 μL) of protein (10 mg/mL in 1 mM TCEP, 5% (v/v) glycerol, 10 mM HEPES-NaOH pH 7.5, and 500 mM NaCl) to mother liquor containing 21%–27.5% (w/v) PEG 1500 and 0.1 M MMT pH 5.75–6.25. Crystals were obtained after 1 week and were soaked at pH 7.0–9.0 with 50 mM ligand present between 12 h and three days. Crystals were flash-vitrified in a solution of mother liquor with 20% (v/v) ethylene glycol and measured at a temperature of 95 K.

## Compound 5

Co-crystals of hWRN in Form E were grown in the presence of **compound 5** by sitting drop vapor diffusion at 4 °C. Prior to drop set-up, hWRN was incubated with 2 mM **compound 5** for 16 h on ice. The drops were set in a 1:1 ratio (0.1 μL: 0.1 μL) of protein mix (10 mg/mL in 1 mM TCEP, 5% (v/v) glycerol, 10 mM HEPES-NaOH pH 7.5, and 500 mM NaCl) to mother liquor containing 20% (w/v) PEG 6000, 0.2 M $CaCl_2$, and 0.1 M MES-NaOH pH 6.0. Crystals were obtained after 3 days and were flash-vitrified in a solution of mother liquor with 20% (v/v) ethylene glycol and measured at a temperature of 95 K.

## Crystal data collection

**AMP-PNP structure.** X-ray diffraction data were collected from crystals under cryogenic conditions at the SWISS LIGHT SOURCE PXII/X10SA (SLS, Villigen, Switzerland) at a wavelength of 1.0000 Å using a Dectris EIGER2 S 16 M detector. The crystals belong to space group C2. For all structures, data were processed using the programs autoPROC (1.1.7)[48], XDS (Jan 31, 2020)[49], and AIMLESS (0.7.7)[50].

## Fragment 1, apo form D, fragment 2, fragment 3, fragment 4, and compound 1

X-ray diffractions were collected from crystals under cryogenic conditions at the SWISS LIGHT SOURCE PXII/X10SA (SLS, Villigen, Switzerland) at wavelengths between 0.9999 Å to 1.0000 Å using a Dectris EIGER2 S 16 M detector. The crystals belong to space group $P2_12_12_1$.

## Compound 3

X-ray diffraction data were collected from crystals under cryogenic conditions at the DIAMOND LIGHT SOURCE IO4 (DLS, Oxford, England) at a wavelength of 0.9537 Å using a Dectris EIGER2 XE 16 M detector. The crystals belong to space group $P2_12_12_1$.

## Compound 5

The X-ray diffractions were collected from complexed crystals under cryogenic conditions at the SWISS LIGHT SOURCE PXII/X10SA (SLS, Villigen, Switzerland) at a wavelength of 0.9999 Å using a Dectris EIGER2 S 16 M detector. The crystals belong to space group I222.

## Compound 6

X-ray diffraction data were collected from complexed crystals under cryogenic conditions at the EUROPEAN SYNCHROTRON RADIATION FACILITY ID23-1 (ESRF, Grenoble, France) at a wavelength of 0.8856 Å using a Dectris EIGER2 CdTe 16 M detector. The crystals belong to space group $P2_12_12_1$.

## Structure determination and refinement

The chosen cut-off criterium for all structures is I/σI ≥ 1.2 in the last shell with reflexes equally distributed over all shells. All datasets have been checked for twinning and anisotropy using Phenix.Xtriage (1.21.1–4487)[51]. No indication for twinning is observable in any dataset. The datasets with **compound 5** and AMP-PNP are moderately anisotropic. Processing of these datasets with STARANISO (from autoPROC 1.1.7)[52] and refinement against these data did not improve the electron

density map, and thus, all structures were refined against isotropically processed data.

Crystal structures of hWRN were determined by molecular replacement using Phaser (from CCP4 7.1.015 and 8.0.005)[53]. Proprietary models have been used as search models. For hWRN in distinct conformations, it was crucial to replace the RecA1 and the RecA2 + Zinc-binding domain independently of each other for the first replacement. For all structures, model building and refinement were performed using Coot (0.9.8.91)[54] and the CCP4 software package (7.1.015, 8.0.004, 8.0.005, and 8.0.016)[55], respectively. Compound geometrical restraints were prepared using CORINA (4.4.0 0026 12.08.2021)[56], and refinements were carried out with REFMAC (5.8.0267 and 5.8.0352)[57]. Figures were prepared using PyMOL (2.5.5)[34]. Crystal structures of hWRN were determined by molecular replacement using Phaser (from CCP4 7.1.015 and 8.0.005)[53]. Model quality was assessed, among others, with MolProbity (from CCP4 7.1.015 and 8.0.005)[58]. Proprietary models have been used as search models. For hWRN in distinct conformations, it was crucial to replace the RecA1 and the RecA2 + Zinc-binding domain independently of each other for the first replacement.

Data collection and refinement statistics for all structures can be found in Table SI 3.

### Reporting summary

Further information on research design is available in the Nature Portfolio Reporting Summary linked to this article.

## Data availability

The final coordinates of WRN have been deposited in the PDB with accession codes 9MJS (AMP-PNP), 9MJT (apo Form D), 9MJU (**fragment 1**), 9MJV (**fragment 2**), 9MJW (**fragment 3**), 9MJX (**fragment 4**), 9MJY (**compound 1**), 9MJZ (**compound 3**), 9MK0 (**compound 5**), and 9MK1 (**compound 6**). Published structures described in this manuscript include 6YHR, 8PFO, and 7GQU. All data are provided in the figures, tables, and supplemental information files associated with the manuscript. Compound characterization data is provided in supplementary Data 1. Source data are provided with this paper.

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

## Acknowledgments

We would like to thank the staff of the ESRF and EMBL Grenoble for assistance and support in using beamline ID23-1. We would like to thank Diamond Light Source and the staff of beamline IO4 for assistance with crystal testing and data collection. We acknowledge the Paul Scherrer Institut, Villigen, Switzerland, for the provision of synchrotron radiation beamtime at beamline PXII/X10SA of the SLS. This work was supported by Merck Sharp & Dohme LLC, a subsidiary of Merck & Co., Inc., Rahway, NJ, USA.

## Author contributions

R.L.P., M.M., J.S., A.B.V., M.M.R., R.J.B., M.K., H.B.M., M.S.M., K.M., J.L.M., M.J.T., S.T., J.D.H., and D.F.W. **designed the research;** R.L.P., M.M., J.S., J.R., A.B.V., A.V.B., M.M.R., X.C., J.H., Z.H., M.J.T., H.B.M., K.M., and M.Z. **performed experiments;** R.L.P., M.M., J.S., J.R., A.B.V., A.V.B., M.M.R., R.J.B., M.K., H.B.M., M.S.M., K.M., M.J.T., D.J.K., J.D.H., and D.F.W. **analyzed results;** R.L.P., M.M., J.S., A.B.V., M.M.R., R.J.B., M.K., A.S., D.J.K., D.G.M., S.B.G., and D.F.W. **wrote and edited the manuscript.**

## Competing interests

All authors are employees or former employees of Merck Sharp & Dohme LLC, a subsidiary of Merck & Co., Inc., Rahway, NJ, USA, and may hold stock or stock options in Merck & Co., Inc., Rahway, NJ, USA.
