## [Transparent Peer Review File · Nature Communications]

WRN structural flexibility showcased through fragment-based lead discovery of inhibitors

Corresponding Author: Dr Sandra Gabelli

Version 0:

Reviewer comments:

Reviewer #1

(Remarks to the Author)

The manuscript by Palte et al. presents a fragment-based approach to identify ligands that bind to the Werner helicase (WRN). WRN is a validated but challenging target for the development of certain cancers. Previous efforts to identify inhibitors of WRN using high throughput screening (HTS) have been hampered by the finding that many compounds that show activity in the functional assays used to identify hits, have failed to validate and show activity in cell-based assays. A central hypothesis of the current work is that by screening fragments using biophysical binding assays, validating their binding using X-ray crystallography and applying a structure-guided approach to fragment elaboration, it may be possible to circumvent some of the problems encountered in the HTS assays.

The authors describe two separate hit-finding approaches – one based on NMR experiments and the other using SPR. Each yielded hit fragments that were validated as binders using X-ray crystallography. The crystallographic analysis provided some insight into the challenges presented by WRN as a target – with multiple different conformations of the protein being observed. The authors conclude that the conformational changes observed in the structural data are relevant in the biological activity of WRN and present snapshots of the structural flexibility that exists in the protein. Ultimately, fragments that bound at distinct allosteric sites on WRN were identified, and fragments that bound at one of these sites were selected for further elaboration. This led to compounds with low micromolar activity in a biochemical assay.

Overall, this is an interesting story and presents a nice example of the application of a biophysical and structural screening cascade against an extremely challenging target. In my opinion it would be of interest to a wide audience. However, I believe that there are some key omissions from the data as presented that need to be addressed. There are also several minor suggestions that I believe would strengthen the manuscript.

Major points.

1. Lack of biophysical data in the manuscript files.

The SPR data presented for fragment 3 and fragment 4 show evidence of significant overbinding. The dose-response data are essentially linear up to the maximum tested concentration of 1 mM – at which point the occupancy is already > 100%. This sort of behaviour is not uncommon for weakly binding fragments – but if it were to persist with higher affinity compounds it would lead me to question the reliability of the SPR assay. However, there are no SPR data presented for the higher affinity compounds. I would suggest that SPR data for the compounds showing affinity in the biochemical assay are necessary to support the central hypothesis of the study that the biophysical assays present a means to weed out badly behaved compounds.

2. Compound characterisation

Many of the purified intermediates lack NMR. There is no ¹³C-NMR data presented for any compound. There is no HRMS or purity data. It is unclear how many of the compounds are literature compounds. If they are then a statement that the ¹H-NMR is consistent with literature and a reference is required. If not, proper characterisation must be reported.

Minor points

The authors show two examples of fragments that were identified from the NMR screen, including concentration-dependent broadening of ¹⁹F resonances in the CPMG spectra as well as evidence of binding in ¹H-NMR STD, CPMG and WaterLOGSY. They make the point that binding in multiple experiments increases confidence in the identification of binders. In the SPR assay, data are presented for fragment 3 and fragment 4. Both show evidence of overbinding in the SPR assay with almost linear responses up to the maximum tested concentration of 1 mM. Fragment 3 is validated as a hit in the NMR assay whereas fragment 4 is not. Both give crystal structures. Given the central hypothesis of the work I suggest that this

warrants comment. Where does the degradation product in the X-ray come from? Is it observed in the stock of the fragment? If not, is it somehow formed due to irradiation at the synchrotron?

I found the order of the figures confusing. The first panel in Figure 1 shows the strategy for chemical elaboration. Figure 2 shows structures and density for fragments and elaborated compounds. Some of the structural data that underpin the elaboration strategy are not presented until Figure 5.

The abstract contains abbreviations that are not defined or explained until later in the text. Likewise, PARP, BRCA1/2 are used at the beginning of the introduction in a manner that I suggest makes it difficult for a non-expert in the field to follow.

ML216 is mentioned in the introduction – but the context of the sentence is not provided until the discussion section.

Druggability is used without a definition of what it mean in the introduction and in the selection of fragment 1 for further work.

P5 – the sentence beginning “internal structure of WRN” may be missing something?

The confidence of the authors in the suitability of the construct is not really explained.

Why is this structure-guided HTE? It seems to be a pretty standard protocol of compound enumeration and docking.

Solubility > 250 mM? In what? How is purity assessed?

How is non-saturable binding assessed from single concentration SPR experiments?

Final assay volume of 20 mL?

Hill (hill) slope.

There are several instances where capitalisation, sub/superscripts appear to be incorrectly formatted

The reference format is inconsistent as are the citation markers

Several of the ESI-MS peaks are quoted to one decimal place with the final digit as 0. Eg. 422.0 – where this number is inconsistent with the expected mass. Is this really the observed mass (with an error)?

There are a number of 19F couplings that I would have expected to see in the 1H NMR that are not reported.

Compound 46 seems to have an in-house code in the experimental.

Reviewer #2

(Remarks to the Author)

Manuscript NCOMMS-25-07608, reviewed by Markus Rudolph and Moritz Classen

The manuscript by Palte et al. describes the discovery and initial SAR of a number of allosteric WRN inhibitors and binders, found by conformation-agnostic pure binding assays. Fragment screens were conducted using SPR and (19F-)NMR, a nice alternative to previous HTS using enzymatic activities. Structural biology showed that the same allosteric site as discussed earlier in the Baltgalvis and Ferretti studies was found, plus additional binder locations. Molecules at these positions, however, need development into inhibitors. The best inhibitor has a comparatively large IC50 of 1.2 microM, which indicates a low affinity for a SAR-derived compound. A novel WRN conformation is present in the crystal structures presented.

Given the fact that no helicase-inhibitor has yet achieved approval, these results are significant to the field and warrant publication. The manuscript would benefit from more comparison with RNA helicases, as for this structurally-related class of non-processive helicases there ample literature on conformational flexibility. For the flexibly-linked RNA helicases it is clear that the conformations "trapped" in crystals are very versatile and may or may not be populated in solution. Domain rotations of "up to 180°" are to be expected in the absence of conformational restrictions. Nonetheless, any conformation that binds an inhibitor, is worthwhile investigating.

A drawback of the manuscript compared to earlier WRN inhibitor studies is the lack of cellular or in vivo data. From a biophysical standpoint, the data appear overall sound and are not over-interpreted. The NMR and SPR experiments are conducted well, and conclusions are drawn cautiously, e.g. by categorizing the SPR traces into different classes. A pity that kinetic information could not be extracted from the SPR data, as a small koff value is often indicative for a good inhibitor. The analysis of binding interfaces gleaned from crystal structures, however, appears too speculative and is not backed up by experimental (e.g. mutational) data. We acknowledge that the authors depict 2Fo-Fc electron density, but more informative to judge the quality of the ligand binding poses are omit maps, as Fourier coefficients for these are usually not deposited in the PDB.

Minor points:

The Methods would benefit from some more details as in our opinion the experiments cannot be reproduced with the information give - see the annotated .pdf files. At times lab jargon is prominent, formatting style is inhomogeneous and some Methods parts have poor written style.

Reviewer #3

(Remarks to the Author)

Reviewer #4

(Remarks to the Author)

Reviewer #5

(Remarks to the Author)

This paper provides a textbook quality hit-to-lead story using fragment-based drug discovery approaches for a challenging target, WRN. The hit-finding strategy and data are very well executed and explained, especially in enumerating the different fragment binding profiles for NMR and SPR. I really liked the rigorous clarity and honesty of this section. The structural data is solid and will be a valuable contribution to the field, providing researchers and the industry with new potential starting points for hit-to-lead efforts on an important target. Certainly, the work merits publication.

However, I'm not convinced this paper is of high enough impact for Nature in its current form. There are a few limitations with the work. First, given the recent literature, the novelty of the allosteric pocket is no longer a major selling point. Second, the reported potency of their compounds is limited in low uM range and no cellular activity is reported. Third, no mechanistic or selectivity data is reported for the compounds. Finally, while new and interesting conformational states of WRN are reported, likely highlighting the dynamics/plasticity of the protein, it's not clear if these findings have biological relevance or if they are merely crystal packing artifacts. If the authors can address some of these concerns, then it would strengthen the case for publication in a high impact journal like Nature Comm. Otherwise, I would suggest a more specialized journal may be more appropriate.

Major Gaps

(1) In terms of data presentation, the authors should really include a SAR summary table in the main figures with the key data summarized for the fragments and compounds. The key data is scattered around in various supplementary figures/tables and is too hard to track, analyze, and compare. A summary table will greatly simplify the key findings for the readership. Format can be something like this:

Compound ID; Chemical Structure; MW & LE; NMR binder (or KD); SPR binder (or KD); Docking Score; EC50 WRN; EC50 BLM; PDB Code

1
2
ETC

(2) One of the major challenges of a fragment-based screening campaign is distinguishing true hits from non-specific aggregators. Please elaborate in the manuscript what was done to remove bad actors and where you incorporated this into the workflow. Was solubility a concern? Please include a brief sentence or two describing how the Merck fragment libraries are curated in terms of properties (MW, solubility, etc). (I see some description in the methods, but I think this point is important enough to elevate to main text).

(3) I was surprised that there is no counter-screening / selectivity data presented in this manuscript. Please include relevant counter-screen data for BLM or whatever helicase you used to assess selectivity.

(4) The "structure guided high throughput experimentation" section (page 6 results) isn't explained well enough. The readership will be very interested to know how Merck, with its state-of-the-art resources, is using computational methods to streamline discovery. Please elaborate this section and methods explaining minimally how the model was validated and a figure showing the predicted docking activities and how those docking scores compared to experimental data. Are we to understand that of 17 compounds predicted by the algorithm, only 2 showed activity (11% success rate and 89% false positive)? This is about what I'd expect and I do believe the transparency and honesty around the persistent challenges with docking is very useful to point out and for the industry to acknowledge with all the current hype around AI.

(5) The mechanistic characterization work is non-existent. Are the compounds competitive with respect to ATP or DNA? Do they show any evidence of cooperativity as was observed with VVD-133214? Novartis published a ATP-gamma-S displacement assay. Do the fragments show any activity in this assay? Please review the existing literature on MOA analyses. These are very easy assays to run and there's no reason to not include those data to better characterize these compounds. Especially for a Nature audience.

(6) The claim about the second bound molecule being a fragment 4 degradation product seems very unlikely. An alternative interpretation of the data could be that a molecule of malonate is bound (a component of MMT crystallization buffer). Can the authors please try modeling malonate into the density and see if this could explain the electron density? Otherwise, please provide relevant stability data on fragment 4 in their crystallization buffer to support their claim.

(7) Please provide some text on whether or not the compounds have any cellular activity. You can use the Novartis compound or Vividion compounds as positive controls. Please address any lack of cellular activity in the text as potency limited or properties limited?

(8) I do like the presentation of the different conformations of WRN. But it's difficult to know if these have any biological relevance. Have the authors done any cryo-EM studies to try and understand at a population level the relative disposition of the two helicase domains in a more native state? Have the authors done any cryo-EM in the presence of DNA? Have the authors done any crystallography in the presence of DNA? Any supporting data would greatly bolster their claims and make their findings more impactful.

Minor

(1) Please move Figure 1 and the electron density Figure 2 to the supplement. Please move the bioNMR and SPR workflows to the main figures. Please move representative NMR data like S3 A&B and representative SPR data to main

figures. These are the key data that informed which fragments to call out as hits. This is more important than the chemistry schematic, which is less informative and barely discussed in the main text.

(2) Did the authors profile their compounds in the published DNA unwinding assay? Please report these data for comparison to their ATPase data.

(3) For pymol figures, please repeat rendering with no shadows. For Figure 4, is hot pink the best color scheme? As a general rule, structure figures are easier to understand when they leverage color and contrast to help highlight the important message. For instance, if the compound binding mode is the "message", color the compound with a brighter color than the protein so it will better stand out. As currently rendered, the compounds are camouflaged with the same color scheme as the rest of the protein. For example, in Figure 3A, GMPPNP adenosine ring is gray and the compound is blue and are so perfectly camouflaged with the protein as to be almost invisible. This would be the time to use a hot pink rendering on the compound so it can be seen against the blue protein background. Same comments apply to Figure 4A and D. Don't camouflage what's important!

(4) For Figure 3 D and F, is it possible to find the same view to show the same set of residues for each compound, fragment 1 and fragment 2? It looks like the figures were made in isolation or autogenerated with a script without considering the importance of cross-comparison.

(5) Page 5: I think there's a missing word: "internal structure of WRN..." Please edit the sentence.

(6) Please provide some further explanation in the text why fragment 1 was advanced and not the others? The current explanation about favorable SAR is a bit vague.

(7) The authors performed a fragment growing strategy in their hit-to-lead progression. Did the authors ever consider a fragment linking approach? This would necessitate screening for a second fragment binder in the presence of one of their preferred/validated fragments? Some comments in the text about their strategic decisions around growth vs linking would be nice.

Version 2:

Reviewer comments:

Reviewer #1

(Remarks to the Author)

As I highlighted in my original review, I believe that the work presented in this manuscript will be of general interest. My original comments on the significance and data interpretation stand, and I comment here on the revisions based on the initial reviews.

The authors appear to have attempted to address most of the comments received on review. I remain highly confused by the apparent reluctance to show the primary data on which the manuscript is based. This is a manuscript that seeks to demonstrate the advantages presented by biophysical screening cascades in identifying chemical matter that binds to challenging targets such as this and is developable into higher affinity compounds. And yet only three examples of the SPR data are shown, none of which are in the main manuscript. Two of these are for very weak compounds and are not ideal examples of SPR binding characterisation for the reasons that I detailed in my previous review. The final compound shows saturable binding with clear dose-response. However, the R_{max} in the dataset for compound 4 (MW 437) is 7 RU. In contrast, the two fragments (MW 246 & 217, respectively) have R_{max} of ~18 and 16 RU and display significant overbinding. The methods do not report any differences in the loading of the protein for the SPR assay - and perhaps the authors have overlooked updating the methods section to reflect inclusion of the new data. However, I would have expected that compounds 2, 3, 4, 5, 6 and 7 should all show saturable binding in the SPR assay - and a demonstration of consistent responses and reproducible protein activity on the surface would lend a great deal more confidence in the robustness of the assay. I would regard the robustness of the SPR assay as important as this seems to be one of the major messages of the paper.

Notwithstanding the addition of the HRMS data, I remain a little surprised by the lack of compound characterisation data. (No ^{13}C NMR, no stated purities). This places quite an emphasis on analysis of the 1H NMR data. The authors state in the response letter that the 1H NMR spectra have been provided - but I could not see them in the resubmission package. Moreover, there are several compounds where there is extensive overlap in the 1H NMR data, and where intensities and couplings could not be resolved or are not reported in the manuscript. I fear that based on my experience I do not share the authors level of confidence that this represents unequivocal characterisation of all compounds.

For example - in the characterisation of compound 4, the three protons on the fluorophenol ring are reported as 7.13 (dd, J 11.2, 8.3, 1H) 6.85 (dd J 8.5, 2.0), and 6.74 (ddd J 8.3, 4.2, 2.1). Presumably the 11.2 Hz is a ^{19}F - 1H coupling, the ddd is has two 1H - 1H couplings and a 4J to the ^{19}F , which makes the dd the proton next to the OH. This would mean that there is a 4J coupling of 8.5 Hz between the proton and the ^{19}F . That is very unusual - particularly given that the other 4J coupling has a more typical value of 4.2 Hz. That is not to say that the spectra have been reported incorrectly - just that in the absence of other data, I would be very uncomfortable reporting this as unequivocal characterisation of the compound's identity.

My other minor gripe is the description of the additional density observed in the crystals structure of fragment 4 as a degradation product. When it is first described the conclusion is that it is probably a degradation product as there is nothing that would be expected to be present in the crystallisation condition that is consistent with the density. And therefore one possible explanation is that it is a degradation product. Thereafter it is simply referred to as "the degradation product". I don't think that the authors have shown definitively that this is what it is - and therefore I suggest that it is inappropriate to refer to it

as such.

Reviewer #2

(Remarks to the Author)

While evading the time-consuming task to provide additional mutational/cellular data, the authors did a good job in responding to the issues raised, imo.

Reviewer #3

(Remarks to the Author)

Reviewer #5

(Remarks to the Author)

This revised version is much improved and overall, I'm supportive for publication. Congratulations on the nice work!

Minor points, in the merged PDF, lines 313-323 appear to be intended for rebuttal letter and not the main text.

Regarding fragment 4 degradation, in my experience, interpreting odd bits of density is always subjective. The language of the main text is appropriately qualified. But I would encourage the authors to also look at the QC of their DMSO stock solution (LCMS) to confirm if such a degradation fragment exists and/or look at the stability of their fragment in crystallization buffer. This would be helpful supporting evidence for their claim. In the absence of such data, one must consider other possibilities like a partially ordered / partial occupancy fragment 4. Or one could imagine a component of the buffer that has more than one conformation that is creating a confusing density blur. My general inclination is to be more conservative in these situations given the uncertainties.

Version 3:

Reviewer comments:

Reviewer #1

(Remarks to the Author)

I am satisfied that the authors have suitably addressed all of the comments in my prior reviews

Reviewer #5

(Remarks to the Author)

The authors have addressed my concerns and I have no further comments. I am supportive for publication.

REVIEWER COMMENTS

Reviewer #1 (Remarks to the Author):

The manuscript by Palte et al. presents a fragment-based approach to identify
ligands that bind to the Werner helicase (WRN). WRN is a validated but challenging target
for the development of certain cancers. Previous efforts to identify inhibitors of WRN using
high throughput screening (HTS) have been hampered by the finding that many compounds
that show activity in the functional assays used to identify hits, have failed to validate and
show activity in cell-based assays. A central hypothesis of the current work is that by
screening fragments using biophysical binding assays, validating their binding using X-ray
crystallography and applying a structure-guided approach to fragment elaboration, it may
be possible to circumvent some of the problems encountered in the HTS assays.

The authors describe two separate hit-finding approaches – one based on NMR
experiments and the other using SPR. Each yielded hit fragments that were validated as
binders using X-ray crystallography. The crystallographic analysis provided some insight
into the challenges presented by WRN as a target – with multiple different conformations
of the protein being observed. The authors conclude that the conformational changes
observed in the structural data are relevant in the biological activity of WRN and present
snapshots of the structural flexibility that exists in the protein. Ultimately, fragments that
bound at distinct allosteric sites on WRN were identified, and fragments that bound at one
of these sites were selected for further elaboration. This led to compounds with low
micromolar activity in a biochemical assay.

Overall, this is an interesting story and presents a nice example of the application of
a biophysical and structural screening cascade against an extremely challenging target. In
my opinion it would be of interest to a wide audience. However, I believe that there are
some key omissions from the data as presented that need to be addressed. There are also
several minor suggestions that I believe would strengthen the manuscript.

We want to thank the reviewer for the comments. We find them to be constructive
critiques of the science we share. We think the comments were addressed as follow
and hope that the improved manuscript will be re-considered for publication.

Major points.**34 1. Lack of biophysical data in the manuscript files.**

The SPR data presented for fragment 3 and fragment 4 show evidence of significant
overbinding. The dose-response data are essentially linear up to the maximum tested
concentration of 1 mM – at which point the occupancy is already > 100%. This sort of
behavior is not uncommon for weakly binding fragments – but if it were to persist with
higher affinity compounds it would lead me to question the reliability of the SPR assay.
However, there are no SPR data presented for the higher affinity compounds. I would
suggest that SPR data for the compounds showing affinity in the biochemical assay are
necessary to support the central hypothesis of the study that the biophysical assays
present a means to weed out badly behaved compounds.

**Response:** To address reviewer concerns around final chemical matter we have
added SPR data for the higher affinity compound 4 (Figure SI 9). We have also added a
statement about compound affinity into the main text.

2. Compound characterization. Many of the purified intermediates lack NMR. There is no
¹³C-NMR data presented for any compound. There is no HRMS or purity data. It is unclear
how many of the compounds are literature compounds. If they are then a statement that
the ¹H-NMR is consistent with literature and a reference is required. If not, proper
characterization must be reported.

**Response:** We have included the HRMS data for all compounds, including both
fragments and newly synthesized final compounds. Additionally, we have provided the ¹H
NMR spectra. These data, along with the X-ray structures, unequivocally characterize the
compounds and confirm their identities.

**Minor points**

1) The authors show two examples of fragments that were identified from the NMR screen,
including concentration-dependent broadening of ¹⁹F resonances in the CPMG spectra as
well as evidence of binding in ¹H-NMR STD, CPMG and WaterLOGSY. They make the point
that binding in multiple experiments increases confidence in the identification of binders.
In the SPR assay, data are presented for fragment 3 and fragment 4. Both show evidence of
overbinding in the SPR assay with almost linear responses up to the maximum tested
concentration of 1 mM. Fragment 3 is validated as a hit in the NMR assay whereas
fragment 4 is not. Both give crystal structures. Given the central hypothesis of the work I
suggest that this warrants comment.

**Response:** We appreciate the reviewer's comment to explain further the
relationship of our hypothesis and our protocol. The optimal output of a fragment screen is
chemical matter with structural information, enabling a structure-based drug design
paradigm. We observed, like others when conducting fragment screens, that chemical
matter with the largest degree of overlap between parallel methods increases the chances
of success for its structure determination by X-ray crystallography. In our NMR screening
paradigm and follow-up, we often prioritize fragments from Rank 1. We do not consider SPR
as a pass/fail criterion as our ¹⁹F library includes compounds with solubility around 250
μ M, making it difficult to acquire good quality SPR data for compounds in the 1 mM affinity
range.

Regarding the SPR results, we purposefully selected compounds with differential
bioNMR profiles to test if we could identify different binding sites. Indeed, we found
different binding sites, but they are shallower and potentially more challenging to advance
chemical matter in them. As this is the first project in which we performed this type of
approach, we don't feel comfortable making more general statements about the
implementation of this workflow to find different binding sites.

2) Where does the degradation product in the X-ray come from? Is it observed in the stock
of the fragment? If not, is it somehow formed due to irradiation at the synchrotron?

**Response:** Apo WRN was crystallized using a buffer comprised of a combination of
MES, DL-malic acid, and Tris, referred to here as MMT. These crystals were subjected to
soaking experiments, including one with Fragment 4. It is important to note that this buffer
does not include malonate. We attempted to model malate in the corresponding density
peak; however, this did not fully account for the density observed, and a positive difference
density remains after refinement. Additionally, none of the other components of the
crystallization buffer could be successfully fitted to the identified electron density.
While one might speculate that malate from the buffer or another fragment/ligand could be
present in the apo structure, this is not supported by the data. Given these factors, we
conclude that malate is not bound, but rather a degradation product from Fragment 4 may
be present. Since the compound was soaked at a concentration of 50 mM, there is a
possibility that trace contaminants are present at sufficient concentrations to occupy the
binding pocket. We do not attribute this to irradiation effects during data collection, as the
density is consistently clear for one of the compounds across the crystal, making it
improbable that only one of the two molecules is subject to degradation from the X-ray
exposure. We have included density images here for the reviewer's further reference
below.

Within the main text we have updated the paragraph containing this information to
read: "One region of density was assigned to a single molecule of **fragment 4** with high
confidence, while the second region was interpreted as a possible degradation product of a
second **fragment 4** molecule rather than buffer components. This interpretation was based on
best fit to initial Fo-Fc density (Figure 4) and was modeled accordingly (Figures 6D-6F)."

The electron density map after 10 cycles of refinement in Refmac.
Density malonate 2Fo-Fc map at 1σ . This molecule is too small and not present in
the crystallization buffer

Density malate 2Fo-Fc map at 1σ . The fit of malate into the density is worse than
from the degradation product of fragment 4

3) I found the order of the figures confusing. The first panel in Figure 1 shows the strategy
for chemical elaboration. Figure 2 shows structures and density for fragments and
elaborated compounds. Some of the structural data that underpin the elaboration strategy
are not presented until Figure 5.

**Response:** We appreciate the comment of the reviewer. Since figures are numbered
and shown in order of their description within the manuscript, we will take suggestions of
what reordering would help with the presentation and readability. At another reviewer's
request, we have also moved NMR and SPR figures into the main text as Figures 1 and 2,
respectively, so please be aware that overall numbering has changed.

4) The abstract contains abbreviations that are not defined or explained until later in the
text.

**Response:** The definitions for MSI-H (Microsatellite instability-high) and MMRd
(Mismatch repair deficiency) have been added to the abstract.

5) Likewise, PARP, BRCA1/2 are used at the beginning of the introduction in a manner that I
suggest makes it difficult for a non-expert in the field to follow.

**Response:** The sentence has been reworded for clarity as requested. It now reads,
"Notable examples of this type of drug therapy are PARP inhibitors for the treatment of
cancers with mutations in BRCA1/2 which are DNA damage repair proteins^{4,5}."

6) ML216 is mentioned in the introduction – but the context of the sentence is not provided
until the discussion section.

**Response:** The sentence was modified to address the reviewer suggestion: "This
lack of success was recently very thoroughly evaluated by Hauser et al. showing that non-
specific interactions can be attributed to impurities in the test samples (ML216) or covalent
labeling of the WRN protein (NSC compounds)."

7) Druggability is used without a definition of what it means in the introduction and in the
selection of fragment 1 for further work.

**Response:** We have edited these sections to more clearly describe the qualities we
were looking for in a binding site. The text now reads: "In addition to these challenges, the
ATP binding site in helicases is predicted to be less likely to bind a drug-like molecule than

enzymes like kinases, complicating the search for potent ATP-competitive inhibitors²³.
These considerations indicate that allosteric inhibition of WRN might facilitate the
identification of selective leads among RECQ helicases. “

8) The sentence beginning “internal structure of WRN” may be missing something?

**Response:** Indeed, the sentence was incomplete upon initial submission. This has
been remedied, and the wording of the sentence has changed slightly to further enhance
the understanding. It now reads: “Our initial structure of wild type human WRN (500-942)
complexed with AMP-PNP exhibits a root mean square deviation (RMSD) of 0.52 Å (Ca
backbone atoms)³⁴ when aligned with PDB ID 6YHR, a longer WRN construct consisting of
residues 517-1093,⁷ despite lacking the HDRC domain.”

9) The confidence of the authors in the suitability of the construct is not really explained.

**Response:** We have added additional descriptions to this section in this regard. Our
initial overlay with the at-the-time only WRN structure available in the PDB lent us
confidence in our construct’s suitability for SAR development. This was further bolstered
by the use of similarly – though not identically – truncated WRN constructs in both 2024
papers regarding clinically-relevant WRN inhibitors where they drive SAR/SBDD with these
constructs to develop inhibitors with in vivo and cellular activity.

10) Why is this structure-guided HTE? It seems to be a pretty standard protocol of
compound enumeration and docking.

**Response:** We thank the reviewer for bringing this to our attention to provide a better
subtitle. We have modified the heading of the paragraph in the main text. Please see
below the extracted paragraph from the main text.

**“Structure guided evolution of fragment 1 series leads to functional inhibition**
**and discovery of a new WRN form**

*We advanced the development of **fragment 1** by maintaining its interactions with key*
*residues Arg732, Tyr849, and Arg910, while also incorporating insights gained from the X-*
*ray structure of **fragment 2** complexed with WRN. Notably, the carboxylic acid of **fragment***
***2** forms an important ionic interaction with Arg857, which informed the evolution of*
***fragment 1**. To design ligands that combined the structural features present in **fragment 1***
*and **fragment 2**, we generated a virtual library by coupling trifluoroaniline with various*
*proprietary acids from our corporate collection. Virtual ligands were assessed based on*
*their docked poses, focusing on key interactions identified in the crystal structures of WRN*
*with bound **fragment 1** and **fragment 2**. Specifically, we looked for hydrogen bonds with*
*Arg732 from **fragment 1**, hydrogen bonds with Arg857 from **fragment 2**, and p-stacking*
*interactions with Tyr849 observed in both fragments. A total of 17 compounds that satisfied*
*all three criteria were selected for synthesis. Among these, we identified several promising*
*molecules, including **compound 1** and its thiophene analog, **compound 2**”*

11) Solubility > 250 mM? In what? How is purity assessed?

Response: We thank the reviewer for bringing this to our attention. The solubility was corrected to 250 μ M. The solubility was assessed by 1 H NMR in the presence of 25 μ M TSP in the standard PBS buffer. LC-MS assessed the purity, and fragments with a purity of more than 90% were selected to be library members. In addition, the library is routinely tested for compound stability, and new mixtures are prepared when compound degradation is observed.

12) How is non-saturable binding assessed from single concentration SPR experiments?

Response: To screen the fragments by SPR, we had to limit our data acquisition to single-point concentration. At this step, we resorted to clustering compounds based on the sensorgram profile. Indeed, the analysis is not perfect as it is only a single point but using the ratio of observed binding RU (at the end of each test fragment injection) and the theoretical maximum occupancy based on a 1:1 binding model allowed us to deprioritize bad actors (fragments with non-saturable /super-stoichiometric profile) efficiently.

13) Final assay volume of 20 mL?

Response: We thank the reviewer for bringing this to our attention. This should be in micromolar, not millimolar. We have edited the document to use the proper micro symbol " μ "

14) Hill (hill) slope.

Response: The capitalization has been corrected.

15) There are several instances where capitalization, sub/superscripts appear to be incorrectly formatted.

Response: We thank you for your attention to detail. We have read through the manuscript carefully and believe we have made all necessary changes.

16) The reference format is inconsistent as are the citation markers.

Response: We appreciate you bringing this to our attention. We have carefully checked for consistency and made all necessary changes.

17) Several of the ESI-MS peaks are quoted to one decimal place with the final digit as 0. Eg. 422.0 – where this number is inconsistent with the expected mass. Is this really the observed mass (with an error)?

Response: We thank the reviewer for allowing us to clarify. These are observed mass with an error. These are from the low resolution mass spectrometry. We have added HRMS data for all of the fragments and final compounds.

18) There are a number of 19 F couplings that I would have expected to see in the 1 H NMR that are not reported.

**Response:** We thank the reviewer for allowing us to further explain. Fluorine in
"CF₃" is 4 bonds away from the nearest aromatic C-H. In our opinion the lack of observed
coupling between F and H is likely due to this distance. Furthermore, we have added the
¹H NMR spectra of fragment 1, where this coupling is also absent.

19) Compound 46 seems to have an in-house code in the experimental.

**Response:** We would like to thank the reviewer for pointing out this typo. We have
renumbered the compounds and fixed this typo.

**Reviewer #2 (Remarks to the Author):**

Manuscript NCOMMS-25-07608, reviewed by Markus Rudolph and Moritz Classen

The manuscript by Palte et al. describes the discovery and initial SAR of a number of
allosteric WRN inhibitors and binders, found by conformation-agnostic pure binding
assays. Fragment screens were conducted using SPR and (¹⁹F-)NMR, a nice alternative to
previous HTS using enzymatic activities. Structural biology showed that the same allosteric
site as discussed earlier in the Baltgalvis and Ferretti studies was found, plus additional
binder locations. Molecules at these positions, however, need development into inhibitors.
The best inhibitor has a comparatively large IC₅₀ of 1.2 microM, which indicates a low
affinity for a SAR-derived compound. A novel WRN conformation is present in the crystal
structures presented.

1) Given the fact that no helicase-inhibitor has yet achieved approval, these results
are significant to the field and warrant publication. The manuscript would benefit from
more comparison with RNA helicases, as for this structurally-related class of non-
processive helicases there ample literature on conformational flexibility. For the flexibly-
linked RNA helicases it is clear that the conformations "trapped" in crystals are very
versatile and may or may not be populated in solution. Domain rotations of "up to 180°" are
to be expected in the absence of conformational restrictions. Nonetheless, any
conformation that binds an inhibitor, is worthwhile investigating.

**Response:** Thank you for your suggestions. We have included additional
information and citations around the known flexible nature of helicases in general and have

highlighted both the SF2 superfamily and RecQ family. We do find significance in the full
180° rotation of these domains as, to our knowledge, no structures apart from recent
studies have shown this significant of a conformational change, especially within the RecQ
family of helicases and BLM in particular. There are several papers discussing a flexible
aromatic loop connecting the HDRC helicase domain to the winged helix (WH) domain of
RecQ, but the flexible linker we describe here is between the two ATPase domains
themselves. Reviewer 5 brought up similar points, and we have included additional
comments in our response there.

2) A drawback of the manuscript compared to earlier WRN inhibitor studies is the lack of
cellular or in vivo data. From a biophysical standpoint, the data appear overall sound and
are not over-interpreted. The NMR and SPR experiments are conducted well, and
conclusions are drawn cautiously, e.g. by categorizing the SPR traces into different
classes. A pity that kinetic information could not be extracted from the SPR data, as a
small koff value is often indicative for a good inhibitor.

**Response:** We thank the reviewer for the opportunity to explain our point of view.
The lack of cellular activity reported for our evolved Fragment Series was also pointed out
by Reviewer #5. We thank Reviewer #2 for recognizing the quality of our biophysical work.
While we now have included the SPR sensorgrams for the highest binding affinity
compound 4 from our work in Figure SI 9 (Kd = 1.2 uM), kinetic information cannot be
derived as correctly stated by the Reviewer. The goal of our biophysical work was to
provide multiple non-covalent fragment screening hits as potential starting points for lead
discovery against this challenging drug target, which is similar in scope to the excellent
paper recently reported in Nature Communications by Hommel, et al. (Discovery of a
selective and biologically active low-molecular weight antagonist of human interleukin-1β |
Nature Communications). Given the status of our reported series, cellular or in vivo data
are out of scope.

3) The analysis of binding interfaces gleaned from crystal structures, however, appears too
speculative and is not backed up by experimental (e.g. mutational) data. We acknowledge
that the authors depict 2Fo-Fc electron density, but more informative to judge the quality
of the ligand binding poses are omit maps, as Fourier coefficients for these are usually not
deposited in the PDB.

**Response:** We have updated all density images to show the Fo-Fc “omit map”
densities for ligands, including the addition of a Fo-Fc map for the AMP-PNP-bound
structure. Regarding the binding interfaces, the data statistics from these structures
including the resolution and B-factors corroborates our interpretations. In our opinion the
general binding sites of compounds analyzed during compound lead discovery efforts are
not addressed with mutational data.

We have added a paragraph in results that address why not mutational data:

Although inclusion of mutagenesis experiments would provide valuable insights into the
importance of improving binding affinity through ionic interactions. In our experience, this
approach is particularly beneficial when working with highly potent compounds, allowing for the

modification or removal of undesired functionalities to enhance drug-like properties or mitigate
off-target liabilities. However, in the relatively weaker affinity space we are currently exploring,
mutagenesis of key residues linked to potency improvements via productive interactions may not
not be detected. Consequently, this approach was not prioritized at this phase of our
optimization. Nevertheless, three orthologous methods, X-ray structures, biophysical and
biochemical data, along with the presence of trifluorophenyl in both our fragment and HRO761,
provide substantial evidence supporting the validity of these compounds for further
development.

Minor points:

1) The Methods would benefit from some more details as in our opinion the experiments
cannot be reproduced with the information give - see the annotated .pdf files. At times lab
jargon is prominent, formatting style is inhomogeneous, and some Methods parts have
poor written style.

**Response:** We appreciate the request for more information on our methods and
have included additional information throughout the methods sections. We have edited
the manuscript to remove any jargon and increase the quality of writing overall.

Reviewer #3 (Remarks to the Author):

I co-reviewed this manuscript with one of the reviewers who provided the listed reports.
This is part of the Nature Communications initiative to facilitate training in peer review and
to provide appropriate recognition for Early Career Researchers who co-review
manuscripts.

Reviewer #4 (Remarks to the Author):

I co-reviewed this manuscript with one of the reviewers who provided the listed reports.
This is part of the Nature Communications initiative to facilitate training in peer review and
to provide appropriate recognition for Early Career Researchers who co-review
manuscripts.

Reviewer #5 (Remarks to the Author):

This paper provides a textbook quality hit-to-lead story using fragment-based drug
discovery approaches for a challenging target, WRN. The hit-finding strategy and data are
very well executed and explained, especially in enumerating the different fragment binding
profiles for NMR and SPR. I really liked the rigorous clarity and honesty of this section. The

structural data is solid and will be a valuable contribution to the field, providing
researchers and the industry with new potential starting points for hit-to-lead efforts on an
important target. Certainly, the work merits publication.

However, I'm not convinced this paper is of high enough impact for Nature in its current
form. There are a few limitations with the work.

1) First, given the recent literature, the novelty of the allosteric pocket is no longer a major
selling point.

**Response:** We are delighted that the reviewer highlights that the manuscript is a
textbook quality hit-to-lead story using fragment-based drug discovery approaches for a
challenging target, WRN, and so merits publication. We also appreciate the opportunity to
elaborate why the data presented is a good fit for publication in Nature Communications.
In our view, the novelty of our work lies not only in the discovery of an allosteric pocket but
also in our ability to identify multiple validated non-covalent hits, a feat that has proven
challenging for others in the field (doi: 10.1002/cmdc.202300613). As the reviewer pointed
out previous work identified a single hit (doi.org/10.1038/s41586-024-07350-y), such
scarcity is not a robust path forward. Moreover, our fragment-based lead discovery
workflow has revealed several binding sites, reinforcing the manuscript's innovative
contributions to the field. Therefore, the novelty of this work is significant in at least three
key aspects.

2) Second, the reported potency of their compounds is limited in low uM range and no
cellular activity is reported.

**Response:** The lack of cellular activity reported for our evolved Fragment Series was
also pointed out by Reviewer #2. Please refer to our answer to that Reviewer *vide supra* for
additional details. Briefly, the workflow and data presented in the manuscript is a fragment-
based lead discovery effort which by definition would identify small chemical fragments
that may bind only weakly to the target, before growing them or combining them to produce
a lead with a higher affinity. In the early phase of FBLD, libraries with a few thousand
compounds with molecular weights of around 200 Da may be screened, and millimolar
affinities can be considered useful. In this range of low affinity binding, low microM,
assessing cellular activity is not conducive. In this sense this request is beyond the scope
of the paper. Nature communication articles included in the Fragment collection are also
examples that lack cellular activity ([https://www.nature.com/articles/s41467-023-41190-](https://www.nature.com/articles/s41467-023-41190-0#Sec10)
[0#Sec10](https://www.nature.com/articles/s41467-023-41190-0#Sec10))

3) Third, no mechanistic or selectivity data is reported for the compounds.

**Response:** We appreciate the opportunity offered by the reviewer to include the
selectivity data. We have incorporated the selectivity data into Table SI 2.

As we explained before, attempting to answer mechanistic questions with compounds in
the μM affinity range is not a customary workflow, and requested assays were not
established.

4) Finally, while new and interesting conformational states of WRN are reported, likely
highlighting the dynamics/plasticity of the protein, it's not clear if these findings have
biological relevance or if they are merely crystal packing rs.

**Response:** We acknowledge that the identification of protein conformations and
interactions may be influenced by crystallization condition. On the other hand, for WRN,
previous work by Ferretti (<https://doi.org/10.1038/s41586-024-07350-y>) and Balgatvis
(<https://doi.org/10.1038/s41586-024-07318-y>) on FORMS A-D have shown the biological
relevance. Furthermore, given that compounds that bind to the allosteric pocket in Forms B-
D were developed into clinical inhibitors, we are confident that these conformations are
populated and relevant in biological contexts and could contribute to enzyme inhibition.
Form E, which is being described for the first time here, further emphasizes the flexibility of
the D1-D2 linker region. While we cannot definitively prove the presence of this form in a
genuine biological environment, based on our understanding of helicase flexibility and the
differences seen with WRN as compared to BLM, for instance, we hypothesize that this
conformation is accessible. Furthermore, we agree with reviewer #2 whom states that: 'any
conformation trapped is worthwhile investigating'.

If the authors can address some of these concerns, then it would strengthen the case for
publication in a high impact journal like Nature Comm. Otherwise, I would suggest a more
specialized journal may be more appropriate.

**Response:** We appreciate the opportunity given to us to address the concerns of
the reviewers and hope to have strengthened the case for publication in Nature
Communications.

Major Gaps

(1) In terms of data presentation, the authors should really include a SAR summary table in
the main figures with the key data summarized for the fragments and compounds. The key
data is scattered around in various supplementary figures/tables and is too hard to track,
analyze, and compare. A summary table will greatly simplify the key findings for the
readership. Format can be something like this:

Compound ID; Chemical Structure; MW & LE; NMR binder (or KD); SPR binder (or KD);
Docking Score; EC50 WRN; EC50 BLM; PDB Code

**Response:** We thank the reviewer for this suggestion that allows a better
presentation of our data. To accommodate Reviewer's suggestions, we modified Table SI 2
and included the structure, MW & LE; IC50 WRN; IC50 BLM; and PDB Code. Please see the
table below. We have discussed more about docking scores in answer #4 following this
and we refer the reviewer to read more there.

Compound	Structure	MW (LE)	ATPase WRN IC ₅₀ (μM)	ATPase BLM IC ₅₀ (μM)	PDB code
Fragment 1		203.1 (NA)	>100	>100	9MJU
Fragment 2		274.2 (NA)	>100	78*	9MJV
Fragment 3		246.7 (NA)	>100	>100	9MJW
Fragment 4		217.2 (NA)	>100	>100	9MJX
Compound 1		313.2 (0.23)	>100	73.4	9MJY
Compound 2		329.3 (0.26)	68.3 ± 2.0	3.7*	
Compound 3		421.4 (0.25)	9.8 ± 4.0	22.8	9MJZ
Compound 4		437.3 (0.28)	1.2 ± 0.7	>100	
Compound 5		515.4 (0.18)	24.6 ± 20.7	28.3	9MK0
Compound 6		565.6 (0.19)	4.2 ± 1.1	4.5	9MK1
Compound 7		594.6 (0.14)	23.7 ± 0.6	14.8	

* Compounds with Hill slope ≥ 2.

(2) One of the major challenges of a fragment-based screening campaign is distinguishing
true hits from non-specific aggregators. Please elaborate in the manuscript what was done
to remove bad actors and where you incorporated this into the workflow. Was solubility a
concern? Please include a brief sentence or two describing how the Merck fragment
libraries are curated in terms of properties (MW, solubility, etc). (I see some description in
the methods, but I think this point is important enough to elevate to main text).

**Response:** We incorporated the suggestions in the first results subsection (entitled
**Fragment-Based Screens identify allosteric binders to WRN**) by adding the following text: “To
identify new binding sites on WRN, we employed our FBLD platform which relies on
screening our internal fragment collections that were a priori curated to only include
compounds with sufficient solubility and purity.” To the Reviewer: To decrease the number
of false positives we prioritized compounds that showed binding with the combination of
468 ^{19}F T₂ CPMG, ^1H T₂ CPMG, STD, and waterLOGSY NMR experiments.

(3) I was surprised that there is no counter-screening / selectivity data presented in this
manuscript. Please include relevant counter-screen data for BLM or whatever helicase you
used to assess selectivity.

**Response:** As we mentioned in our earlier response for point #3 under limitations,
addressing mechanistic and selectivity data, we appreciated the opportunity offered by the
reviewer to include the selectivity data. We have incorporated the selectivity data to Table
SI 2.

(4) The “structure guided high throughput experimentation” section (page 6 results) isn’t
explained well enough. The readership will be very interested to know how Merck, with its
state-of-the-art resources, is using computational methods to streamline discovery.
Please elaborate this section and methods explaining minimally how the model was
validated and a figure showing the predicted docking activities and how those docking
scores compared to experimental data. Are we to understand that of 17 compounds
predicted by the algorithm, only 2 showed activity (11% success rate and 89% false
positive)? This is about what I’d expect, and I do believe the transparency and honesty
around the persistent challenges with docking is very useful to point out and for the
industry to acknowledge with all the current hype around AI.

**Response:** We thank the reviewer for allowing us to expand and clarify. Following
the suggestion we changed the title from, ‘structure guided high throughput
experimentation’, to ‘Structure guided evolution of fragment 1 series leads to functional
inhibition and discovery of a new WRN form’. The new text attempts to provide clarity
regarding our methods for selecting compounds. For example, instead of using docking
scores to predict binding or inhibitory activity, our focus was on identifying which
geometries would facilitate the polar interactions observed in the crystal structures of
fragments 1 and 2 bound to WRN. We concur with the reviewer opinion that an 11% hit rate
aligns with the anticipated success for this type of approach.

(5) The mechanistic characterization work is non-existent. Are the compounds competitive
with respect to ATP or DNA? Do they show any evidence of cooperativity as was observed
with VVD-133214? Novartis published an ATP-gamma-S displacement assay. Do the
fragments show any activity in this assay? Please review the existing literature on MOA
analyses. These are very easy assays to run and there's no reason to not include those
data to better characterize these compounds. Especially for a Nature audience.

**Response:** is a significant potency right shift for WRN inhibitors when going from
ATPase inhibition to DNA unwinding inhibition. This suggests that a high degree of ATPase
inhibition is required to significantly reduce observed DNA unwinding rates for WRN.
Consequently, the threshold for helicase inhibition is not easily overcome for initial
fragment screening derived small molecule hits. This consideration also applies to
accurate execution of MOA studies, where weaker compounds with High IC50s present
challenges in the execution of precise and informative MOA studies, which require use of
much higher compound concentrations close to or exceeding their solubility limits. As a
discovery campaign progresses and potency of the molecules continue to improve down to
hundreds of nM, MOA studies become more feasible, and translatability to the helicase
assay more accessible. We have incorporated these considerations into our discovery
plan. We did, however, conduct selectivity analysis for our compounds, comparing WRN
and BLM ATPase activity. This data has been included in the manuscript (Table S2) to
facilitate better interpretation of the compounds.

(6) The claim about the second bound molecule being a fragment 4 degradation product
seems very unlikely. An alternative interpretation of the data could be that a molecule of
malonate is bound (a component of MMT crystallization buffer). Can the authors please try
modeling malonate into the density and see if this could explain the electron density?
Otherwise, please provide relevant stability data on fragment 4 in their crystallization
buffer to support their claim.

**Response:** We refer the reviewer to see our detailed response to Reviewer #1 who
shared similar concerns regarding the degradation product of fragment 4. In short, we
provide detailed information for reviewers to look over during the review process that we
believe substantiates our hypothesis around a degradation product and have
simultaneously edited the manuscript to address this concern. (from above) In our opinion
since, Apo WRN was crystallized using a buffer comprised of a combination of MES, DL-
malic acid, and Tris, referred to here as MMT. These crystals were subjected to soaking
experiments, including one with Fragment 4. We would like to highlight that the buffer has
Malic acid and not malonate acid. So we attempted to model malate in the corresponding
density peak; however, this did not fully account for the density observed, and a positive
difference density remains after refinement. Additionally, none of the other components of
the crystallization buffer could be successfully fitted to the identified electron density.
While one might speculate that malate from the buffer or another fragment/ligand could be
present in the apo structure, this is not supported by the data. Given these factors, we
conclude that malate is not bound, but rather a degradation product from Fragment 4 may

be present. Since the compound was soaked at a concentration of 50 mM, there is a
possibility that trace contaminants are present at sufficient concentrations to occupy the
binding pocket. We do not attribute this to irradiation effects during data collection, as the
density is consistently clear for one of the compounds across the crystal, making it
improbable that only one of the two molecules is subject to degradation from the X-ray
exposure. We have included density images here for the reviewer's further reference
below.

Within the main text we have updated the paragraph containing this information to
read: "One region of density was assigned to a single molecule of **fragment 4** with high
confidence, while the second region was interpreted as a possible degradation product of a
second **fragment 4** molecule rather than buffer components. This interpretation was based on
best fit to initial Fo-Fc density (Figure 4) and was modeled accordingly (Figures 6D-6F)."

(7) Please provide some text on whether or not the compounds have any cellular activity.
You can use the Novartis compound or Vividion compounds as positive controls. Please
address any lack of cellular activity in the text as potency limited or properties limited?

**Response:** As this is a FBLD campaign no cellular activity was assessed since it is
assumed that the low molecular weight, low affinity, low potency would not yield
substantive results in a cellular assay.

(8) I do like the presentation of the different conformations of WRN. But it's difficult to
know if these have any biological relevance. Have the authors done any cryo-EM studies to
try and understand at a population level the relative disposition of the two helicase
domains in a more native state? Have the authors done any cryo-EM in the presence of
DNA? Have the authors done any crystallography in the presence of DNA? Any supporting
data would greatly bolster their claims and make their findings more impactful.

**Response:** We acknowledge that the determination of protein conformations and
interactions may be influenced by crystallization artifacts. Given the development of
clinical inhibitors that bind to the allosteric pocket in Forms B-D, we are confident that
these conformations are relevant in biological contexts and contribute to enzyme
inhibition. Form E, which is being described for the first time here, further emphasizes the
flexibility of the D1-D2 linker region. While we cannot definitively prove the presence of this
form in a biologically relevant environment, based on our understanding of helicase
flexibility and the differences seen with WRN as compared to BLM, for instance, we
hypothesize that this conformation is accessible. We have not conducted cryoEM studies
as they extend beyond the scope of this paper.

To address this within the manuscript itself we have superimposed the public
structure of BLM with bound DNA (PDB 4CGZ) with WRN Form D and WRN Form E in a new
figure in the Supplementary Information, Fig SI 10. We aligned on the D1 and the D2
domains. In all cases the DNA binding region was accessible. We have updated the section
to read: "Alignments of the ATPase D1 and D2 domains of fragment 1 and compound 5 with
the structure of BLM helicase with bound DNA (PDB 4CGZ)⁴⁰ indicate that DNA is likely
able to bind to the two additional conformations of WRN, Form D and Form E, respectively

(Figure SI 10). Consequently, the disruption of the ATP binding pocket or the prevention of
domain rearrangement to achieve an ATP-binding competent conformation are both
plausible mechanisms of inhibition.”

Figure SI 10. Overlay of BLM with WRN-fragment 1 and WRN-compound 5. (A)
Overlay of D1 domains of BLM (white), WRN-fragment 1 (blue,) and WRN-compound 5
(green) structures. (B) Overlay of the same structures on the D2 domain.

Minor

(1) Please move Figure 1 and the electron density Figure 2 to the supplement. Please move
the bioNMR and SPR workflows to the main figures. Please move representative NMR data
like S3 A&B and representative SPR data to main figures. These are the key data that
informed which fragments to call out as hits. This is more important than the chemistry
schematic, which is less informative and barely discussed in the main text.

**Response:** We appreciate the suggestions to move the figures to make the
manuscript better. Key NMR and SPR figures have been moved to the main text as Figures
1 and 2, respectively. The remaining NMR and SPR data do not fit within the figure limit of
the paper and as such are found in the Supplementary Information file. We have decided to
display the Fo-Fc electron density maps of all ligands for transparency, so we have
retained them as Figure 4.

(2) Did the authors profile their compounds in the published DNA unwinding assay? Please
report these data for comparison to their ATPase data.

**Response:** There is a significant potency right shift for WRN inhibitors when going
from ATPase inhibition to DNA unwinding inhibition. Suggestive that a high degree of
ATPase inhibition is necessary to significantly reduce observed DNA unwinding rates for
WRN. As such, the threshold for helicase inhibition is not easily overcome for initial
fragment screening derived small molecule hits. This consideration also applies to

accurate execution of MOA studies, where weaker compounds with High IC50s present
challenges in the execution of precise and informative MOA studies, which require use of
much higher compound concentrations close to or exceeding their solubility limits. As a
discovery campaign progresses and potency of the molecules continue to improve down
to hundreds of nM, MOA studies become viable and translatability to the helicase assay
more accessible; we would have incorporated both into our discovery plan. We did,
however, conduct selectivity analysis for our compounds, comparing WRN and BLM
ATPase activity. This data has been incorporated into the manuscript (Table S2) for better
interpretation of the compounds.

(3) For pymol figures, please repeat rendering with no shadows. For Figure 4, is hot pink the
best color scheme? As a general rule, structure figures are easier to understand when they
leverage color and contrast to help highlight the important message. For instance, if the
compound binding mode is the “message”, color the compound with a brighter color than
the protein so it will better stand out. As currently rendered, the compounds are
camouflaged with the same color scheme as the rest of the protein. For example, in Figure
3A, GMPPNP adenosine ring is gray and the compound is blue and are so perfectly
camouflaged with the protein as to be almost invisible. This would be the time to use a hot
pink rendering on the compound so it can be seen against the blue protein background.
Same comments apply to Figure 4A and D. Don’t camouflage what’s important!

**Response:** Thank you for the advice. We have removed shadows, change colors
and remade most figures for better clarity. We have added ‘boxes’ to the ligands in the
overall structure. Figure numbers have shifted due to the addition of NMR and SPR figures
to the main text. The example noted here is now Figure 6.

(4) For Figure 3 D and F, is it possible to find the same view to show the same set of
residues for each compound, fragment 1 and fragment 2? It looks like the figures were
made in isolation or autogenerated with a script without considering the importance of
cross-comparison.

**Response:** The two figures were originally made to highlight only residues of
importance to that specific ligand, but we agree that highlighting the same residues
between these two structures can add to the story of fragment development. The images
have been updated to reflect this suggestion. Please note that Figure numbers have shifted
with the addition of NMR and SPR figures to the main text. The example noted here is now
Figure 4.

(5) Page 5: I think there’s a missing word: “internal structure of WRN...” Please edit the
sentence.

**Response:** Thank you for your attention to detail. Indeed, the sentence was
incomplete in the initial submission. This has been remedied, and the wording of the
sentence has changed slightly to further enhance the understanding. It now reads “Our
initial structure of wild type human WRN (500-942) complexed with AMP-PNP exhibits a
root mean square deviation (RMSD) of 0.52 Å (Ca backbone atoms)³⁴ when aligned with

PDB ID 6YHR, a longer WRN construct consisting of residues 517-1093,⁷ despite lacking
the HDRC domain.”

(6) Please provide some further explanation in the text why fragment 1 was advanced and
not the others? The current explanation about favorable SAR is a bit vague.

**Response:** We have combined this answer with the one below as they partially
overlap. In prioritizing the chemical matter, we first evaluate the confidence in binding
interaction using biophysical approaches. The second layer of prioritization is x-ray
crystallography, and all compounds that pass that bar undergo SIMS expansion to
deprioritize fragments with steep SAR. In parallel, the merging and growing approaches
may be triggered to advance the fragments. The fragments 1 and 2 growth vectors highly
overlapped, and merging their features allowed us to efficiently identify compounds 1 and
2 that further evolved to higher affinity chemical matter. We did not perform the fragment
linking as we saw a viable path forward with the growing/merging approach. However, if this
was unsuccessful, one could consider another screen for the linking approach.

(7) The authors performed a fragment growing strategy in their hit-to-lead progression. Did
the authors ever consider a fragment linking approach? This would necessitate screening
for a second fragment binder in the presence of one of their preferred/validated
fragments? Some comments in the text about their strategic decisions around growth vs
linking would be nice.

**Response:** The answer was combined with point above.

REVIEWER COMMENTS

Reviewer #1 (Remarks to the Author):

As I highlighted in my original review, I believe that the work presented in this manuscript will be of general interest. My original comments on the significance and data interpretation stand, and I comment here on the revisions based on the initial reviews.

The authors appear to have attempted to address most of the comments received on review.

We thank the reviewer for highlighting the general interest and significance of these data.

I remain highly confused by the apparent reluctance to show the primary data on which the manuscript is based. This is a manuscript that seeks to demonstrate the advantages presented by biophysical screening cascades in identifying chemical matter that binds to challenging targets such as this and is developable into higher affinity compounds. And yet only three examples of the SPR data are shown, none of which are in the main manuscript. Two of these are for very weak compounds and are not ideal examples of SPR binding characterization for the reasons that I detailed in my previous review. The final compound shows saturable binding with clear dose-response. However, the R_{max} in the dataset for compound 4 (MW 437) is 7 RU. In contrast, the two fragments (MW 246 & 217, respectively) have R_{max} of ~18 and 16 RU and display significant overbinding. The methods do not report any differences in the loading of the protein for the SPR assay - and perhaps the authors have overlooked updating the methods section to reflect inclusion of the new data. However, I would have expected that compounds 2, 3, 4, 5, 6 and 7 should all show saturable binding in the SPR assay - and a demonstration of consistent responses and reproducible protein activity on the surface would lend a great deal more confidence in the robustness of the assay. I would regard the robustness of the SPR assay as important as this seems to be one of the major messages of the paper.

Response: Reviewer 1 makes a good point, and as we settled to center the publication around structural data, we pushed biophysical workflows to the second tier. That said, we did routinely evaluate compounds using biophysical approaches to ensure that we observed the same trend of activity/affinity gain with biochemical and biophysical methods. We now include additional data for compounds 2, 3, 4, 5, and 7 (unfortunately, we do not have data for compound 6 and cannot acquire it as we no longer have the material).

Regarding the comment about lower occupancy and method section, the experimental data are now updated, and indeed we observed that for all advanced chemical matter, occupancy plateaued on average around 50-60%. For compound 5, binding to WRN Form E, we could

never even reach 20% occupancy. We also re-acquired data for compounds 3, 4, 5, and 7 at a lower temperature and observed comparable binding profiles and occupancies. Overall, we never saw an increase above 70% (please see the data at 10°C below). In the sensorgrams below, the temperature and occupancy are shown on the top. Please note that there was a slightly lower density of immobilization at 10°C, which is why we see lower RU signal, but higher occupancy.

We added Figure SI 8. to the Supporting Information in this Revision, to show a representative SPR sensorgram, and where applicable an affinity plot, for compounds **2**, **3**, **4**, **5**, and **7**.

Range of Occupancy at 20°C: 29-41%

Range of Occupancy at 10°C: 22-23%

Notwithstanding the addition of the HRMS data, I remain a little surprised by the lack of compound characterization data. (No ^{13}C NMR, no stated purities). This places quite an emphasis on analysis of the ^1H NMR data. The authors state in the response letter that the ^1H NMR spectra have been provided - but I could not see them in the resubmission package. Moreover, there are several compounds where there is extensive overlap in the ^1H NMR data, and where intensities and couplings could not be resolved or are not reported in the manuscript. I fear that based on my experience I do not share the authors level of confidence that this represents unequivocal characterization of all compounds.

For example - in the characterization of compound 4, the three protons on the fluorophenol ring are reported as 7.13 (dd, J 11.2, 8.3, 1H) 6.85 (dd J 8.5, 2.0), and 6.74 (ddd J 8.3, 4.2, 2.1). Presumably the 11.2 Hz is a ^{19}F - ^1H coupling, the ddd is has two ^1H - ^1H couplings and a 4J to the ^{19}F , which makes the dd the proton next to the OH. This would mean that there is a 4J coupling of 8.5 Hz between the proton and the ^{19}F . That is very unusual - particularly given that the other 4J coupling has a more typical value of 4.2 Hz. That is not to say that the spectra have been reported incorrectly - just that in the absence of other data, I would be very uncomfortable reporting this as unequivocal characterisation of the compound's identity.

Response:

We appreciate the thoroughness of the reviewer's examination of our compound characterization work. In the revised manuscript, we are providing ^1H and ^{13}C spectra for all compounds along with estimations of their purities (>95%) in a revised pdf file. Specifically, we include the complete characterization of compound 4, performed using 2D NMR spectroscopy. Copies of the COSY, NOESY, HSQC, and HMBC spectra are provided. All homonuclear and heteronuclear correlations were consistent with the proposed structure. Regarding the concerns about the proton-fluorine couplings in compound 4, we were slightly surprised by the somewhat higher than expected value of the 4-bond $\text{J}(\text{H},\text{F})$ couplings, which are usually smaller than 6 Hz. However, the DFT computations of the J -couplings (B3LYP/6-31+G(d,p), nmr=mixed) indicate that the observed coupling is consistent with the experimental value.

DFT values: $4\text{J}(\text{H}21,\text{F})=8.0$ Hz, $3\text{J}(\text{H}24,\text{F})=9.8$ Hz, $4\text{J}(\text{H},\text{F})=4.0$ Hz

Exper. Values: $4\text{J}(\text{H}21,\text{F})=8.5$ Hz, $3\text{J}(\text{H}24,\text{F})=11.3$ Hz, $4\text{J}(\text{H},\text{F})=4.2$ Hz

A relatively high value of the $4\text{J}(\text{H},\text{F})$ coupling in this case is likely the result of the electron-withdrawing effect of the hydroxyl group positioned between the interacting nuclei.

My other minor gripe is the description of the additional density observed in the crystals structure of fragment 4 as a degradation product. When it is first described the conclusion is that it is probably a degradation product as there is nothing that would be expected to be present in the crystallization condition that is consistent with the density. And therefore one possible explanation is that it is a degradation product. Thereafter it is simply referred to as "the degradation product". I don't think that the authors have shown definitively that this is what it is - and therefore I suggest that it is inappropriate to refer to it as such.

Response:

We have reworded these sentences to better address this ambiguity.
“...indicating the presence of two separate molecules at this site. One region of density was confidently assigned to a single molecule of fragment 4, while the second region was ambiguous. After analyzing all buffer components and compound byproducts, we chose to model a byproduct of the compound into this space, as it best fit the initial Fo-Fc density (Figure 6E). Although we cannot determine the exact molecule occupying this second region of density, it is evident that the pocket can accommodate this additional chemical matter.... “while the carboxy group of the hypothesized byproduct”

And later

“Furthermore, the structure featuring fragment 4 shows a clear density region for fragment 4, along with another density region that we modeled as a ligand byproduct, within the remnants of the ATP pocket.”

Reviewer #2 (Remarks to the Author):

While evading the time-consuming task to provide additional mutational/cellular data, the authors did a good job in responding to the issues raised, imo.

We thank the reviewer for highlighting the effort in addressing the comments.

Reviewer #3 (Remarks to the Author):

Reviewer #5 (Remarks to the Author):

This revised version is much improved and overall, I'm supportive for publication. Congratulations on the nice work!

We thank the reviewer for highlighting the effort in addressing the comments.

Minor points, in the merged PDF, lines 313-323 appear to be intended for rebuttal letter and not the main text.

Response:

We thank this Reviewer for catching this unintended addition to the Main Text. This section was indeed intended only for the rebuttal letter. We have not included it in the new Main Text currently submitted.

Regarding fragment 4 degradation, in my experience, interpreting odd bits of density is always subjective. The language of the main text is appropriately qualified. But I would encourage the authors to also look at the QC of their DMSO stock solution (LCMS) to confirm if such a degradation fragment exists and/or look at the stability of their fragment in crystallization buffer. This would be helpful supporting evidence for their claim. In the absence of such data, one must consider other possibilities like a partially ordered / partial occupancy fragment 4. Or one could imagine a component of the buffer that has more than one conformation that is creating a confusing density blur. My general inclination is to be more conservative in these situations given the uncertainties.

Response:

We agree that the density is ambiguous enough that many different interpretations could be of the same validity. We adjusted the text to touch on this, rewording the first explanation of this density as follows:

“...indicating the presence of two separate molecules at this site. One region of density was confidently assigned to a single molecule of fragment 4, while the second region was ambiguous. After analyzing all buffer components and compound byproducts, we chose to model a byproduct of the compound into this space, as it best fit the initial Fo-Fc density (Figure 6E). Although we cannot determine the exact molecule occupying this second region of density, it is evident that the pocket can accommodate this additional chemical matter.... “while the carboxy group of the hypothesized byproduct”

And later

“Furthermore, the structure featuring fragment 4 shows a clear density region for fragment 4, along with another density region that we modeled as a ligand byproduct, within the remnants of the ATP pocket.”